# Bid maintains mitochondrial cristae structure and function and protects against cardiac disease in an integrative genomics study

Christi T Salisbury-Ruf[1†], Clinton C Bertram[1†], Aurelia Vergeade[2], Daniel S Lark[3], Qiong Shi[4], Marlene L Heberling[5], Niki L Fortune[4], G Donald Okoye[6], W Gray Jerome[7], Quinn S Wells[4], Josh Fessel[4], Javid Moslehi[6], Heidi Chen[8], L Jackson Roberts II[2,4], Olivier Boutaud[2], Eric R Gamazon[9,10]*, Sandra S Zinkel[1,4]*

[1]Department of Cell and Developmental Biology, Vanderbilt University, Nashville, United States; [2]Department of Pharmacology, Vanderbilt University, Nashville, United States; [3]Molecular Physiology and Biophysics, Vanderbilt University, Nashville, United States; [4]Department of Medicine, Vanderbilt University Medical Center, Nashville, United States; [5]Department of Biological Sciences, Vanderbilt University, Nashville, United States; [6]Division of Cardiovascular Medicine and Cardio-oncology Program, Vanderbilt University Medical Center, Nashville, United States; [7]Department of Pathology, Microbiology, and Immunology, Vanderbilt University Medical Center, Nashville, United States; [8]Department of Biostatistics, Vanderbilt University Medical Center, Nashville, United States; [9]Division of Genetic Medicine, Vanderbilt University Medical Center, Nashville, United States; [10]Clare Hall, University of Cambridge, Cambridge, United Kingdom

*For correspondence:
Eric.gamazon@vanderbilt.edu
(ERG);
sandra.zinkel@vanderbilt.edu
(SSZ)

†These authors contributed equally to this work

Competing interests: The authors declare that no competing interests exist.

**Abstract** Bcl-2 family proteins reorganize mitochondrial membranes during apoptosis, to form pores and rearrange cristae. In vitro and in vivo analysis integrated with human genetics reveals a novel homeostatic mitochondrial function for Bcl-2 family protein Bid. Loss of full-length Bid results in apoptosis-independent, irregular cristae with decreased respiration. *Bid-/-* mice display stress-induced myocardial dysfunction and damage. A gene-based approach applied to a biobank, validated in two independent GWAS studies, reveals that decreased genetically determined BID expression associates with myocardial infarction (MI) susceptibility. Patients in the bottom 5% of the expression distribution exhibit >4 fold increased MI risk. Carrier status with nonsynonymous variation in Bid's membrane binding domain, Bid[M148T], associates with MI predisposition. Furthermore, Bid but not Bid[M148T] associates with Mcl-1[Matrix], previously implicated in cristae stability; decreased MCL-1 expression associates with MI. Our results identify a role for Bid in homeostatic mitochondrial cristae reorganization, that we link to human cardiac disease.
DOI: https://doi.org/10.7554/eLife.40907.001

## Introduction

The critical function for Bcl-2 family proteins during apoptosis transpires at the mitochondria and involves remodeling of both the inner and outer mitochondrial membranes to mobilize cytochrome c and release it into the cytosol. In addition to cell death, mitochondrial membranes can reorganize with changes in metabolic conditions (*Hackenbrock, 1966*) (*Mannella, 2006*). Regulation of the inner mitochondrial membrane (IMM) into highly organized loops known as cristae is necessary for a

**eLife digest** Cells contain specialized structures called mitochondria, which help to convert fuel into energy. These tiny energy factories have a unique double membrane, with a smooth outer and a folded inner lining. The folds, called cristae, provide a scaffold for the molecular machinery that produces chemical energy that the cell can use. The cristae are dynamic, and can change shape, condensing to increase energy output.

Mitochondria also play a role in cell death. In certain situations, cristae can widen and release the proteins held within their folds. This can trigger a program of self-destruction in the cell. A family of proteins called Bcl-2 control such a 'programmed cell death' through the release of mitochondrial proteins. Some family members, including a protein called Bid, can reorganize cristae to regulate this cell-death program. When cells die, Bid proteins that had been split move to the mitochondria. But, even when cells are healthy, Bid molecules that are intact are always there, suggesting that this form of the protein may have another purpose.

To investigate this further, Salisbury-Ruf, Bertram et al. used mice with Bid, and mice that lacked the protein. Without Bid, cells – including heart cells – struggled to work properly and used less oxygen than their normal counterparts. A closer look using electron microscopy revealed abnormalities in the cristae. However, adding 'intact' Bid proteins back in to the deficient cells restored them to normal.

Moreover, without Bid, the mice hearts were less able to respond to an increased demand for energy. This decreased their performance and caused the formation of scars in the heart muscle called fibrosis, similar to a pattern observed in human patients following a heart attack.

DNA data from an electronic health record database revealed a link between low levels of Bid genes and heart attack in humans, which was confirmed in further studies. In addition, a specific mutation in the Bid gene was found to affect its ability to regulate the formation of proper cristae.

Combining evidence from mice with human genetics revealed new information about heart diseases. Mitochondrial health may be affected by a combination of specific variations in genes and changes in the Bid protein, which could affect heart attack risk. Understanding more about this association could help to identify and potentially reduce certain risk factors for heart attack.

DOI: https://doi.org/10.7554/eLife.40907.002

multitude of metabolic processes (*Cogliati et al., 2016*)(*Rampelt et al., 2017*). Cristae harbor respiratory chain complexes embedded within and peripheral to the membrane and this tight organization is critical for efficient electron transfer (*Lapuente-Brun et al., 2013*) and cytochrome c sequestration (*Korsmeyer et al., 2000*). Inefficient oxidative phosphorylation due to disruption of the respiratory chain can lead to mitochondrial disease, which range widely in organ systems and severity (*Moslehi et al., 2012*; *Picard et al., 2016*; *Wallace, 2013*).

During apoptosis, the BH3-only protein Bid, is cleaved by caspase-8 (cBid) to facilitate both mitochondrial cristae reorganization (*Cogliati et al., 2013*; *Frezza et al., 2006*; *Scorrano et al., 2002*) and outer membrane permeability (*Gross et al., 1999*; *Li et al., 1998*; *Luo et al., 1998*; *Walensky et al., 2006*; *Wang et al., 1996*). cBid associates with the multidomain Bcl-2 proteins Bax and Bak through its BH3-domain at the outer mitochondrial membrane (OMM), triggering mitochondrial outer membrane pores (MOMP) (*Gross et al., 1999*; *Li et al., 1998*; *Luo et al., 1998*; *Walensky et al., 2006*; *Wang et al., 1996*).

Bid's role in regulating cristae structure has been limited to in vitro studies focusing on isolated mitochondria and cBid. cBid's interaction with the mitochondrial membrane is stabilized in part through an interaction with MTCH2 as well as cBid's membrane binding domain (MBD), consisting of alpha-helices 4,5, and 6 (Tae-Hyoung Kim, Yongge Zhao, Wen-Xing Ding, *Kim et al., 2004*). Alpha-helix-6 partially embeds within the membrane (*Oh et al., 2005*), and has been shown to be necessary for apoptotic cristae reorganization (*Cogliati et al., 2013*).

In addition to its apoptotic function, Bid is also known to be involved in the regulation of other essential cellular processes such as DNA damage and metabolism, acting as rheostat for cell health (Reviewed in *Giménez-Cassina and Danial, 2015*; *Hardwick et al., 2012*; *Zinkel et al., 2006*). Interestingly, full-length Bid can also localize to the mitochondria (*Maryanovich et al., 2012*;

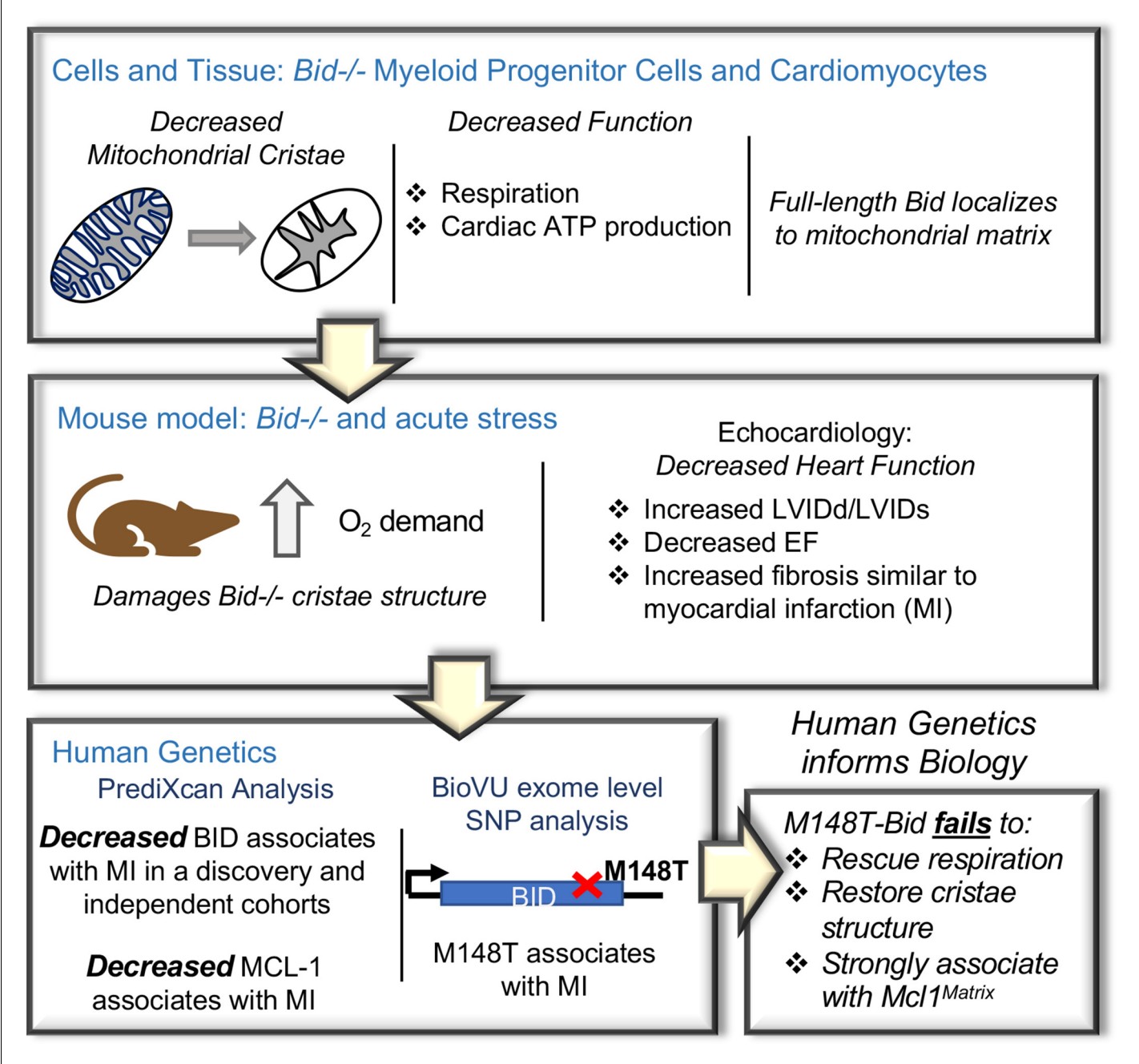

**Figure 1.** An integrated approach combining cells and mice with human genetics uncovers a novel role for Bid in the regulation of mitochondrial cristae. Diagram of the approach used to uncover Bid's novel function in regulating mitochondrial cristae structure. *Bid-/-* myeloid progenitor cell (MPCs) and left ventricular (LV) heart mitochondria have cristae structure abnormalities that result in functional defects. These defects are enhanced under conditions of stress in a *Bid-/-* mouse model. Human genetics analysis using PrediXcan reveals decreased BID gene expression associated with MI and BID exome level variation identifies coding SNP M148T, which is directly linked to Bid's mitochondrial function. This SNP fails to restore cristae structure, respiration, and association with Mcl-1$^{Matrix}$.

DOI: https://doi.org/10.7554/eLife.40907.003

*Wang et al., 2014*). The role for this association and the consequence for mitochondrial function as well as implication for human disease have not been explored.

We reveal a new role for full-length Bid in the regulation of mitochondrial cristae under homeostatic conditions in an approach that integrates cell biology with human genetic studies (*Figure 1*).

We observe that loss of Bid impairs proper cristae formation in the absence of an apoptotic stimulus both in myeloid cells and left ventricular (LV) cardiomyocytes. This function is independent of Bid's caspase-8 cleavage site (D59A), and BH3-domain. We demonstrate decreased respiration in *Bid-/-* cells and decreased respiration coupled with decreased ATP production in LV fibers. These deformations become more pronounced in the heart when it is exposed to various cardiac stressors including Epinephrine and Doxorubicin, in both cases leading to decreased LV function in *Bid-/-* mice. In the case of Epinephrine, these changes correspond to increased cristae damage and fibrosis, phenotypically similar to damage caused by a myocardial infarction (MI) in humans.

Given the known association between mitochondrial dysfunction, especially respiratory chain deficiencies, and heart disorders (*Schwarz et al., 2014*), we use two human genetics approaches to interrogate an association for BID with human cardiac diseases. We first use PrediXcan, which estimates the genetically determined component of gene expression (*Gamazon et al., 2015*; *Gamazon et al., 2018*), applied to a cohort of Vanderbilt University's de-identified genetic database called BioVU (*Roden et al., 2008*). We reveal a highly significant association between decreased BID expression and MI. We also find that patients with the lowest 5% of BID expression have a > 4 fold increase in MI susceptibility. BID's role in cardiac diseases is further validated through an investigation of additional independent cohorts including the large-scale CARDIoGRAMplusC4D GWAS datasets (*Schunkert et al., 2011*)(*Nikpay et al., 2015*). Secondly, using BioVU exome-chip data, we uncover a gene-level association with MI from low-frequency nonsynonymous variation. Of significance, coding single nucleotide polymorphism (SNP) M148T lies within Bid's membrane binding domain (MBD), that includes alpha-helix-6. We then demonstrate that the double Bid mutant Bid$^{BH3/M148T}$ fails to support proper mitochondrial respiratory function or restore cristae in *Bid-/-* cells.

The Bcl-2 family member, Mcl-1, has been shown to localize to the mitochondrial matrix (Mcl-1$^{Matrix}$) and facilitate maintenance of respiratory complexes (*Perciavalle et al., 2012*). We also observe a pool of Bid localized to the matrix and find that while WT Bid can interact with Mcl-1$^{Matrix}$, this matrix association is diminished with Bid$^{M148T}$. Using PrediXcan, we find MCL-1 and MTX-1, a mitochondrial transporter that associates with the mitochondrial contact site and cristae reorganizing complex (MICOS) (*Guarani et al., 2015*) are significantly associated with MI, linking susceptibility to MI to two additional genes involved in cristae regulation.

Our study provides an integrative approach, summarized in *Figure 1*, that spans observations in tissue culture and mice to independent human genetics studies providing direct relevance for our findings in human disease. We shed light on the regulation of mitochondrial cristae and consequently oxidative phosphorylation and reveal an important role for Bid's alpha-helix-6 in regulation of mitochondrial function under homeostatic conditions. Furthermore, this approach provides a model for elucidating previously unrecognized proteins that impact complex genetic diseases.

## Results

### *Bid-/-* cells have a cristae defect that can be rescued with BH3-mutated or D59-mutated Bid

Consistent with a pro-survival function, *Bid-/-* myeloid progenitor cells (MPCs) display decreased growth rates not due to altered proliferation, but instead as a result of decreased viability (p<0.05) (*Figure 2—figure supplement 1a–c*). Given the critical apoptotic role for Bid at the mitochondria, we evaluated mitochondrial structure in *Bid-/-* MPCs by transmission electron microscopy (TEM) (*Figure 2a and b*, *Figure 2—figure supplement 2a*). Compared to *Bid +/+* MPCs, mitochondria in *Bid-/-* MPCs were highly abnormal. Quantitation of the average number of cristae per mitochondrion revealed a significant decrease in the number of cristae in *Bid-/-* MPCs compared to *Bid +/+* MPCs (p<0.0001) (*Figure 2c*). This function is independent of Bid's apoptotic role, as *Bid-/-* MPCs stably expressing Flag-HA-tagged full-length Bid mutated in either in its BH3-domain (FHA-Bid$^{BH3}$) or caspase-8 cleavage site D59 (FHA-Bid$^{D59A}$) could rescue cristae structure (p<0.0001) (*Figure 2c*). Furthermore, quantitation of the area density of mitochondria per cell revealed a slight decrease in density in the *Bid-/-* cells compared to *Bid+/+* cells (p<0.05), while FHA-Bid$^{D59A}$ expressing cells had increased mitochondrial density compared to all other cell lines (*Figure 2d*).

Several groups have reported that cleaved Bid (cBid) reorganizes cristae (*Cogliati et al., 2013*; *Scorrano et al., 2002*) or Bid BH3-peptide narrows cristae junction size (*Yamaguchi et al., 2008*) in

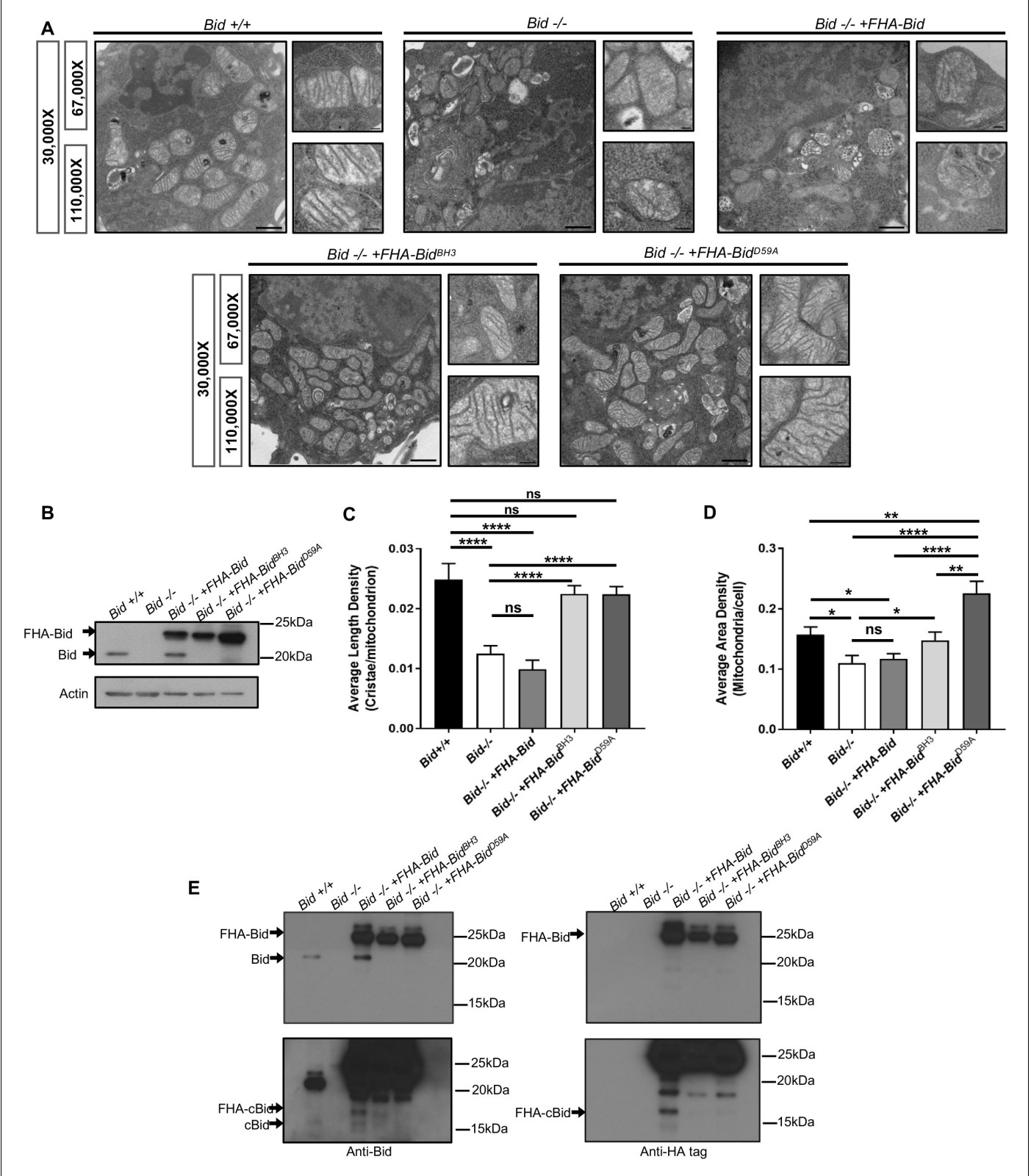

**Figure 2.** The Bcl-2 family protein Bid is required for normal mitochondrial cristae formation, independent from its apoptotic function. (A) Transmission electron microscopy (TEM) of mitochondria from MPC cell lines including: *Bid +/+* (WT), *Bid-/-, Bid-/- + FHA* Bid, *Bid-/- + FHA-Bid*$^{BH3}$ and *Bid-/- + FHA-Bid*$^{D59A}$. Representative images at 30,000X (scale bar = 500 nm), 67,000X and 100,000X magnification (scale bar = 100 nm). Also see *Figure 2—figure supplement 2*. (B) Western blot of expression levels of Bid for the indicated genotypes. Note that full-length Bid is observed in

*Figure 2 continued on next page*

*Figure 2 continued*

Bid-/- + FHA Bid cells due to cleavage of the FlagHA-epitope tag. (**C**) Quantitation of the number of cristae per mitochondria (represented by the average length density) and (**D**) the mitochondrial density per cell (represented by the average area density) of the MPC lines shown in (**A**). A total of 40 images were quantified at 30,000X for each cell line. (**E**) Western blot of Bid (left) and HA-tag (right) indicating increased presence of cleaved Bid (cBid) in Bid-/- + FHA Bid cells (lower blots are darker exposure). FlagHA-tagged expressing cells were loaded for equal Bid expression. P-values were determined by one-way ANOVA (p<0.0001) with unpaired Student's t-test (**C, D**). Error bars indicate ±SEM for all data. ns = not significant, *p<0.05, **p<0.01, ***p<0.001, and ****p<0.0001.

DOI: https://doi.org/10.7554/eLife.40907.004

The following source data and figure supplements are available for figure 2:

**Source data 1.** Data for *Figure 2* and *Figure 2—figure supplement 1*.

DOI: https://doi.org/10.7554/eLife.40907.007

**Figure supplement 1.** Loss of Bid results in increased cell death in the absence of an apoptotic stimulus.

DOI: https://doi.org/10.7554/eLife.40907.005

**Figure supplement 2.** Loss of Bid results in a mitochondrial structural defect not compensated for by upregulation of Bad, Puma, or Bim.

DOI: https://doi.org/10.7554/eLife.40907.006

the presence of isolated mitochondria. Given that myeloid cells have high endogenous protease activity, we anticipated that reintroduction of WT FHA-Bid into *Bid-/-* MPCs by retroviral transduction may not rescue mitochondrial structure (*Figure 2a–d*). Indeed, overexpression of full-length WT Bid (*Bid-/- + FHA Bid*) but not FHA-Bid$^{BH3}$ or FHA-Bid$^{D59A}$ results in the production of endogenous cBid in the absence of a death stimulus (*Figure 2e*). Thus, in a myeloid cell line, we observe that Bid's apoptotic domains must be mutated to fully restore cristae.

We next analyzed expression of other BH3-only apoptotic proteins as we anticipated they may be upregulated in the absence of Bid, and considering the known role of Bim in disassembly of mitochondrial Opa-1 oligomers (*Yamaguchi et al., 2008*). We evaluated *Bid-/-* cellular extracts as well as lysate from left ventricular (LV) cardiac tissue which are highly enriched in mitochondria, and find no compensatory upregulation of Bim, Bad, or Puma to account for the observed loss of cristae structure in *Bid-/-* cells (*Figure 2—figure supplement 2b and c*).

## Full-length Bid localizes to multiple-mitochondrial subcompartments in the absence of cell death

It has previously been shown that full-length Bid can localize to mitochondria in the absence of an apoptotic stimulus (*Maryanovich et al., 2012*; *Wang et al., 2014*). To confirm this result, we first evaluated Bid in subcellular fractions of *Bid-/-* and WT MPCs. We find full-length Bid in a heavy membrane, mitochondrial-enriched fraction absent of cytosolic contamination (*Figure 3a*). We also observe full-length Bid in mitochondria isolated from liver tissue, both in a crude mitochondrial fraction as well as in a Percoll purified fraction (*Figure 3b*).

To determine the submitochondrial localization of full-length Bid, isolated liver mitochondria were treated with Proteinase K (PK) in the presence or absence of SDS. We observe that a pool of Bid remains uncleaved with PK, under conditions in which we observe cleaved Bak (*Figure 3c*), a protein associated with the OMM. Furthermore, we used an osmotic shock approach to separate and enrich for OMM and mitoplast (inner membrane and matrix containing fractions) from isolated liver mitochondria. We find an enrichment of Bid in the mitoplast-containing fraction compared to the OMM (*Figure 3d*). Taken together, the above results suggest that full-length Bid can localize to the mitochondria during non-apoptotic conditions and is found both at the OMM as well as in the mitoplast.

## *Bid-/-* mice have abnormal left ventricular mitochondrial cristae exacerbated by acute cardiac stress

Mitochondria cristae defects in humans can result in severe abnormalities in multiple organ systems, especially the heart (*Brown et al., 2017*; *Meyers et al., 2013*). We were interested to know if *Bid-/-* mice also display cristae abnormalities beyond myeloid cells. TEM of left ventricular tissue isolated from *Bid-/-* mice revealed striking irregularities both in gross mitochondrial organization between myofibrils as well as loss of normal lamellar cristae structure (*Figure 4a*). Specifically, without treatment, *Bid-/-* tissue had overall decreased mitochondrial electron density corresponding to significantly increased cristae width (p<0.0001) (*Figure 4b*).

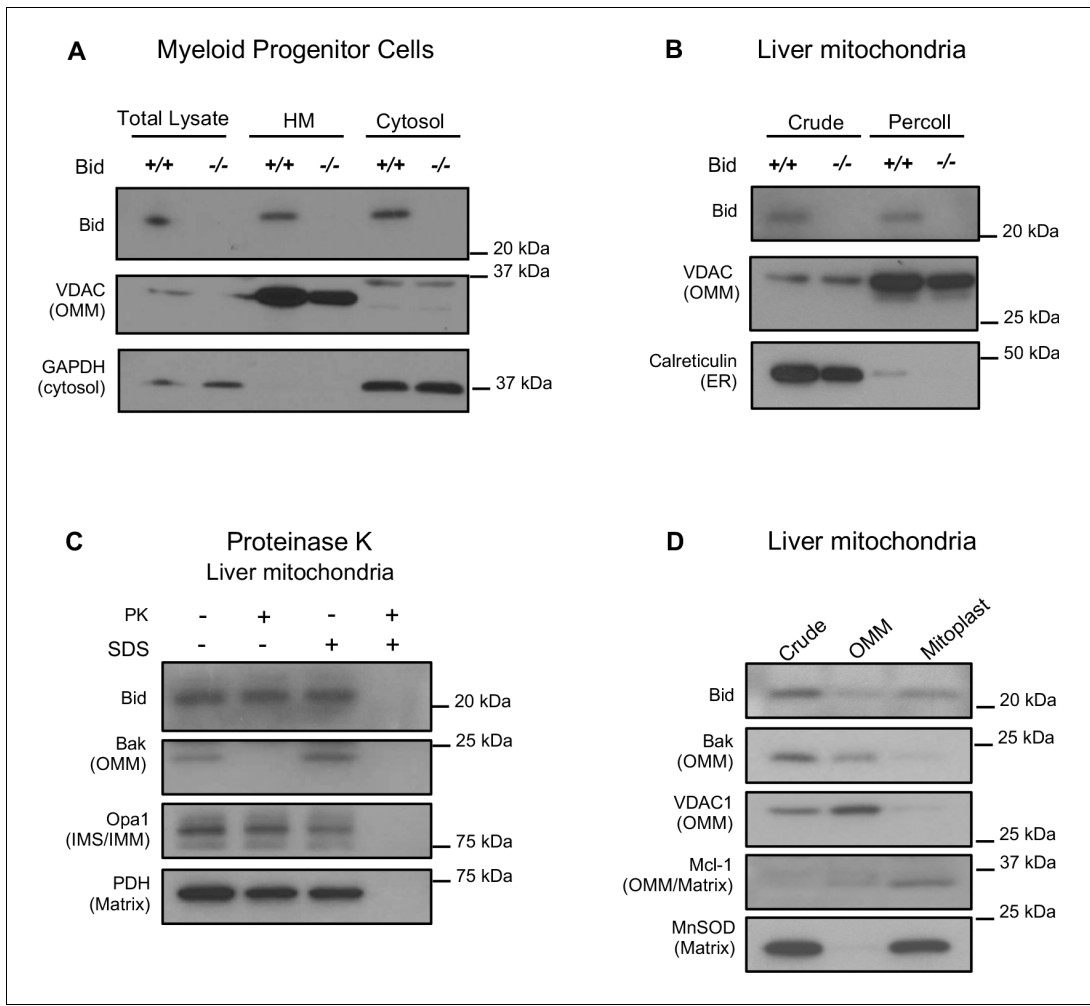

**Figure 3.** Full-length Bid localizes to the mitochondria in the absence of apoptosis and is found within a mitoplast fraction. (**A**) Subcellular fractionation of WT and *Bid-/-* MPCs showing whole cell lysate (WCL), a mitochondrial-containing heavy membrane (HM) (VDAC) and cytosolic fraction (GAPDH). (**B**) Crude and Percoll purified liver mitochondria from WT and *Bid-/-* mice shows the presence of full-length Bid in the purified fraction in the absence of light membrane contamination. (**C**) Proteinase K (PK) treatment of isolated liver mitochondria reveals Bid in a PK resistant fraction. (**D**) Crude liver mitochondria from WT mice were fractionated into OMM and mitoplast (IMM/matrix) containing fractions and probed for OMM and matrix markers. Full-length Bid can be observed in the mitoplast rich fraction. OMM = outer mitochondrial membrane, IMS = inner membrane space, IMM = inner mitochondrial membrane.

DOI: https://doi.org/10.7554/eLife.40907.008

To test how *Bid-/-* mice respond to an acute stress, we used Epinephrine (Epi) to increase the energetic demand on the mitochondria. We assessed both *Bid+/+* and *Bid-/-* mitochondria 18 hr after a dose of 0.5 mg/kg Epi and find that while both *Bid+/+* and *Bid-/-* tissues are damaged, the *Bid-/-* cristae are significantly more deformed ($p<0.0001$) (*Figure 4a and b*). Interestingly, these damaged cristae are structurally similar to mitochondria observed after induction of an acute myocardial infarction (MI) (*Bryant et al., 1958*). Thus, *Bid-/-* mice have a severe cardiac cristae defect that results in increased susceptibility to acute stress-induced damage.

## Acute cardiac stress results in a functional defect in *Bid-/-* mice

To determine whether the mitochondrial cristae defect in *Bid-/-* mice translates to decreased cardiac function, we performed echocardiograms on mice. In the absence of a clear mouse model of heart failure (*Breckenridge, 2010*), we chose Epi as an acute pharmacological stress due to the fact it

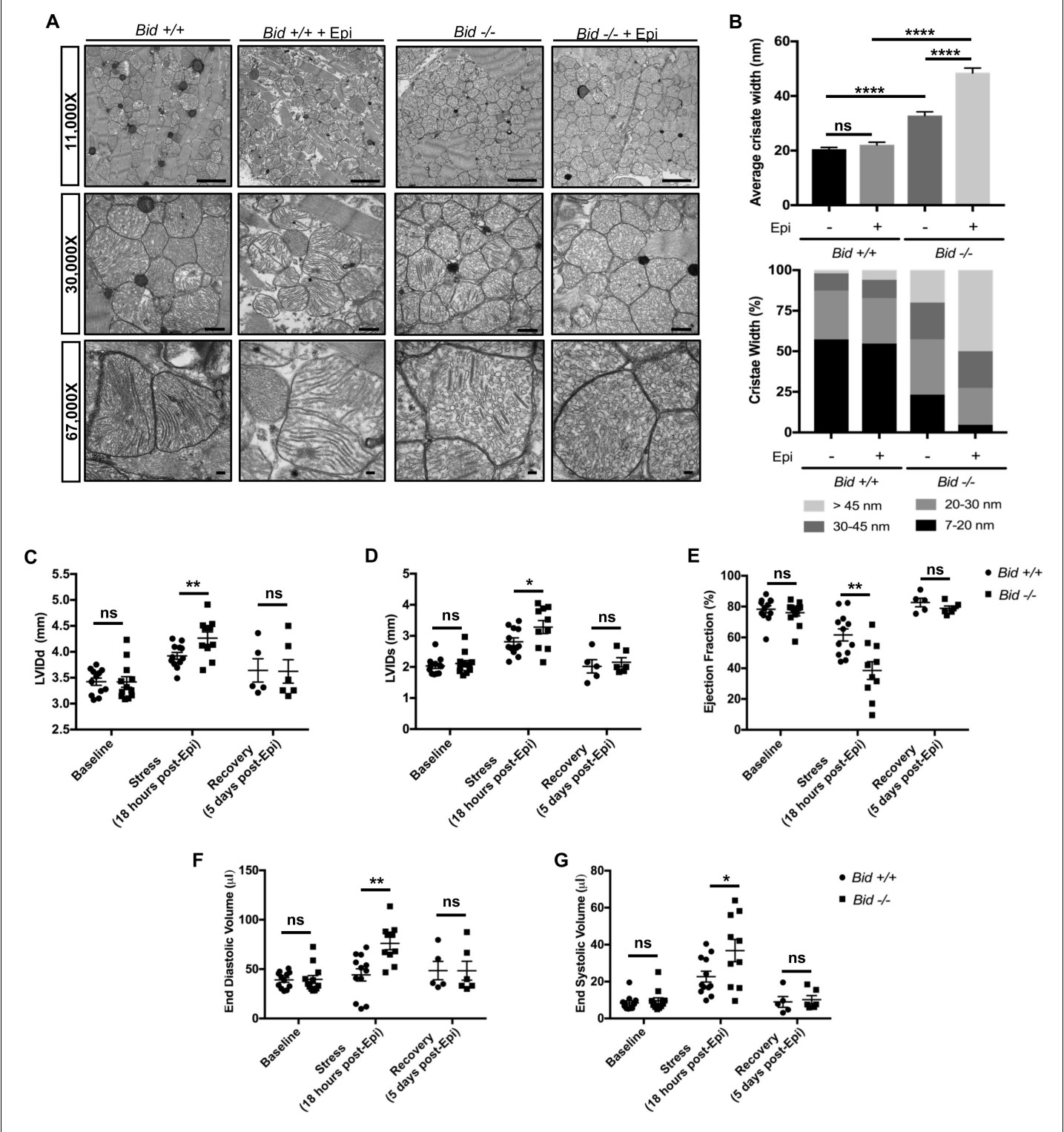

**Figure 4.** Left ventricular cardiomyocytes from *Bid-/-* mice have abnormal cristae, which are structurally and functionally exacerbated with acute Epinephrine stress. (A) Transmission electron microscopy (TEM) of left ventricle cardiomyocyte mitochondria from *Bid +/+* and *Bid-/-* 18 hr with or without 0.5 mg/kg Epinephrine treatment. Representative images at 11,000X (scale bar = 2 μm), 30,000X (scale bar = 500 nm), and 67,000X (scale bar = 100 nm). (B) (Top) Quantitation of average cristae width (nanometers) corresponding to (A). n = 150 cristae per genotype, measured at 67,000X. (Bottom) Percent of cristae corresponding to the indicated widths (nm). (C) Echocardiogram analysis of left-ventricular internal diameter diastole (LVIDd, mm) and (D) LVID systole (LVIDs, mm) of *Bid +/+* and *Bid-/-* mice at the indicated time points. (E) Ejection fraction (%) from *Bid +/+* and *Bid-/-* mice
*Figure 4 continued on next page*

*Figure 4 continued*

without treatment (Baseline), 18 hr after 0.5 mg/kg Epinephrine, and 120 hr post Epinephrine (recovery). (F) End diastolic volume (µl) and (G) End systolic volume (µl) at the indicated time points. n = 12, 12, 5 *Bid +/+* mice and n = 12, 10, 6 for *Bid-/-* mice for baseline, 18 hr, and 120 hr time points, respectively for (C–G). P-values were determined by one-way ANOVA with unpaired Student's t-test (B), and unpaired Student's t-test (C–G). Error bars indicate ± SEM for all data. ns = not significant, *p<0.05, **p<0.01 and ****p<0.0001.
DOI: https://doi.org/10.7554/eLife.40907.009

The following source data and figure supplement are available for figure 4:

**Source data 1.** Data for *Figure 4* and *Figure 4—figure supplement 1*.
DOI: https://doi.org/10.7554/eLife.40907.011
**Figure supplement 1.** *Bid-/-* mice have decreased FS and EF with chronic Doxorubicin.
DOI: https://doi.org/10.7554/eLife.40907.010

causes both a rise in blood pressure with increased left ventricular (LV) afterload as well as increased myocardial contractility (*Goldberg et al., 1960*). This results in maximal oxygen demand with potential to reveal a phenotype driven by mitochondrial dysfunction.

*Bid+/+* and *Bid-/-* mice were evaluated at baseline (without treatment), 18 hr after acute-intraperitoneal (IP) Epi (0.5 mg/kg), and 5 days after Epi treatment to evaluate recovery. Cardiac function does not differ at baseline. However, we find a significant increase in left internal ventricular diameter during diastole (LVIDd) (p<0.01) as well as during systole (LVIDs) (p<0.05) 18 hr after Epi (*Figure 4c and d*). This corresponds to a significant decrease in Ejection Fraction (EF) for *Bid-/-* mice (p<0.01) (*Figure 4e*) and a trend for decreased fractional shortening (FS) (p=0.1564) (*Figure 4—figure supplement 1a*). Furthermore, we also observe an increase in both end diastolic (p<0.01) and end systolic volume (p<0.05) (*Figure 4f and g*) with stress. This is consistent with our findings by EM indicating a decreased ability of *Bid-/-* hearts to maintain proper mitochondrial structure under stress. Decreased LV cardiac function observed in *Bid-/-* mice is phenotypically similar to observations made by echo in patients during the acute phase of MI (LV wall dilation and decreased ejection fraction) (*White et al., 1987*; *Di Bella et al., 2013*). Interestingly, at 5 days post-Epi, both the *Bid+/+* and *Bid-/-* mice had restored cardiac function and we find no difference in heart weights at sacrifice (*Figure 4c–g* and *Figure 4—figure supplement 1b*).

Lastly, we also employed an additional pharmacological myocardial stress in the form of Doxorubicin (Dox) (3 doses of 7.5 mg/kg), a chemotherapy drug with heart mitochondrial toxicity (*Hull et al., 2016*). Dox also resulted in a significant decrease in FS and EF (p<0.01) (*Figure 4—figure supplement 1c and d*) in *Bid-/-* mice. Thus, using two different models, Epinephrine, which directly results in increased oxygen demand, as well as the mitochondrial toxic drug Doxorubicin, we find that Bid plays a role in maintaining LV function under stress.

### *Bid-/-* hearts have increased fibrotic damage after acute stress, similar to post-MI damage observed in human patients

Myocardial fibrosis due to cardiomyocyte remodeling after damage is a prominent sequelae of MI, and directly contributes to loss of cardiac function (*Talman and Ruskoaho, 2016*). To determine the extent of fibrotic damage, we used Masson's trichrome staining and quantitatively evaluated whole heart tissue sections (*Figure 5*). We find that *Bid-/-* tissue has significantly increased fibrosis both at the 18 hr and the recovery time point, 5 days post treatment (p<0.05) (*Figure 5b and c*). Interestingly, WT mice display no increase in fibrosis at 18 hr post-Epi; fibrosis developed in WT hearts at 5 days post-Epi. Thus, *Bid-/-* mice have more fibrosis and increased susceptibility to damage after stress. This result recapitulates the response to cardiomyocyte damage in human MI and suggests that although the *Bid-/-* mice are able to recover functionally, the long-term damage is more severe.

### Loss of Bid results in decreased respiratory complex subunits and ATP synthase dimer activity

To better understand how loss of Bid alters mitochondrial function, we performed proteomics using Multidimensional Protein Identification Technology (MudPIT) on equal concentrations of isolated mitochondrial protein from *Bid +/+* and *Bid-/-* MPCs (*Figure 6—figure supplement 1a*). We identified a total of 3258 proteins that mapped to unique Entrez gene identifiers. Cross referencing our

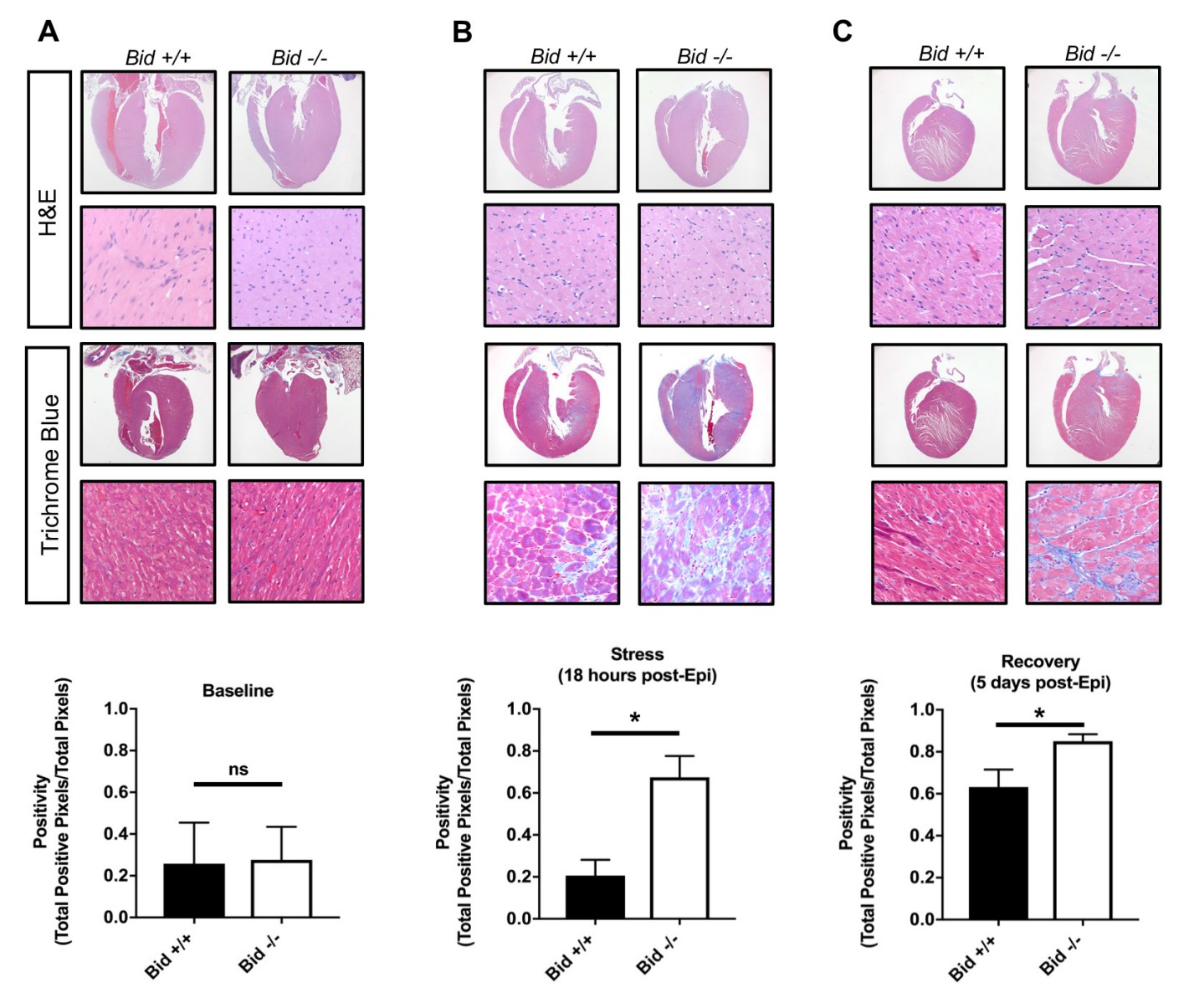

**Figure 5.** Epinephrine stress results in increased fibrotic damage in *Bid-/-* hearts. (A) Representative images of H and E staining of *Bid+/+* and *Bid-/-* hearts (top) and Masson's Trichrome staining (bottom) without treatment. Quantitation of the Trichrome positivity (Total positive pixels/Total pixels), n = 3,3 respectively. (B) H and E and Masson's Trichrome 18 hr after Epinephrine (0.5 mg/kg) with quantitation as in (A), n = 4, 3. (C) H and E and Masson's Trichrome 5 days after Epinephrine (0.5 mg/kg) with quantitation, n = 6,6. P-values were determined by unpaired Student's t-test. Error bars indicate ± SEM for all data. ns = not significant, *p<0.05.
DOI: https://doi.org/10.7554/eLife.40907.012

The following source data is available for figure 5:

**Source data 1.** Data for *Figure 5*.
DOI: https://doi.org/10.7554/eLife.40907.013

hits to the Mouse MitoCarta 2.0 (*Calvo et al., 2016*), we identified 54 significantly different mito-chondrial proteins between the *Bid+/+* and *Bid-/-* samples (*Figure 6—figure supplement 1b and c*).

Our MudPIT results suggested a possible defect in mitochondrial respiratory chain function. To interrogate individual respiratory complexes, we isolated mitochondria from heart tissue of age matched *Bid +/+* (WT) and *Bid-/-* mice. We then resolved digitonin-extracted complexes using gradi-ent Native-PAGE, stained with Coomassie blue, and treated with complex-specific substrates to

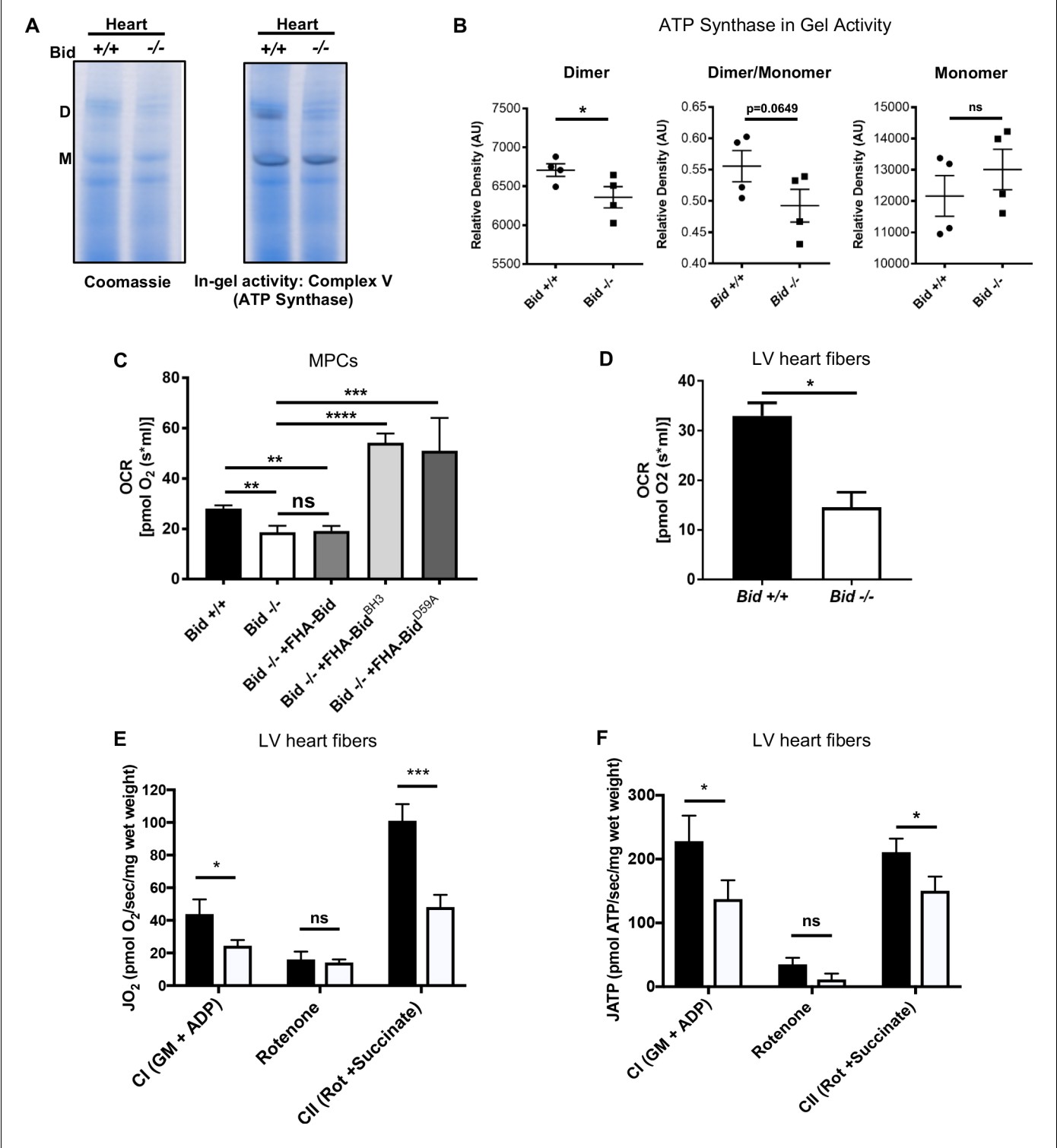

**Figure 6.** Loss of Bid results in decreased ATP-synthase activity and respiration. (**A**) Representative native gel (left) and in-gel activity (IGA) assay (right) for Complex V (ATP synthase) from isolated *Bid +/+* and *Bid-/-* heart mitochondria, D = dimer and M = monomer of ATP synthase. (**B**) Quantitation of IGA assay for heart CV activity as measured by the relative density of indicated dimer and monomer bands (arbitrary units), (n = 4). Also see ***Figure 6—figure supplement 1D–F*** for additional respiratory complex activity analysis. (**C**) Oxygen consumption rate (OCR) was measured in an Oroboros Oxygraph in complete IMDM media on equivalent numbers of indicated cells ($2 \times 10^6$) (n = 7,7,4,3,3 respectively). (**D**) State 3 respiration of saponin permeabilized left ventricle cardiac fibers from *Bid +/+* and *Bid-/-* mouse hearts in MiRO5 respiration medium supplemented with glutamate, malate, and ADP (n = 3). (**E**) Oxygen consumption ($JO_2$) of permeabilized left ventricular cardiac fibers from *Bid+/+* and *Bid-/-* mice in the presence of indicated

*Figure 6 continued on next page*

*Figure 6 continued*

metabolic substrates. G = glutamate, M = malate, CI = Complex I, CII = Complex II, Rot = Rotenone (n = 6,6 respectively). (F) Simultaneous ATP synthesis in presence of metabolic substrates as in (E). P-values determined by unpaired Student's t-test (B), (D), one-way ANOVA (p<0.0001) with unpaired Student's t-test (two-way) for (C), and two-way ANOVA (p<0.01) with unpaired Student's t-test for (E) and (F). Error bars indicate ±SEM for all data. ns = not significant, *p<0.05, **p<0.01,***p<0.001, and ****p<0.0001.

DOI: https://doi.org/10.7554/eLife.40907.014

The following source data and figure supplements are available for figure 6:

**Source data 1.** Data for *Figure 6*, *Figure 6—figure supplement 1*, and *Figure 6—figure supplement 2*.
DOI: https://doi.org/10.7554/eLife.40907.017
**Figure supplement 1.** *Bid-/-* MPC mitochondria have a decreased CI proteins and decreased activity, but no difference in CIV activity.
DOI: https://doi.org/10.7554/eLife.40907.015
**Figure supplement 2.** *Bid-/-* MPCs maintain their membrane potential and do not increased ROS.
DOI: https://doi.org/10.7554/eLife.40907.016

measure enzymatic activity. We observe a decrease in the activity of ATP synthase dimers from *Bid-/-* heart mitochondria (p<0.05) (*Figure 6a and b*), consistent with the known association between dimerization of ATP synthase in cristae loop formation and stabilization of cristae structure (*Hahn et al., 2016*; *Paumard et al., 2002*). Enzymatic activity of additional respiratory complexes and supercomplexes were also evaluated including Complex I (CI) and complex I V (CIV). We observed a significant decrease in the activity of complex I within the SCs and a trend for decreased activity of complex I V containing SCs (*Figure 6—figure supplement 1d,e and f*). Overall, these results are consistent with a role for Bid in maintenance of cristae structure linked to respiratory chain function.

## *Bid-/-* MPCs display decreased respiration

We next measured respiration directly from *Bid-/-* MPCs and LV fibers. Using an Oroboros Oxygraph, we found *Bid-/-* MPCs displayed significantly decreased oxygen consumption rates (OCR) compared to *Bid+/+* cells (p=0.008), consistent with a cristae defect. Respiration could be restored in *Bid-/-* MPCs by re-introduction of FHA-Bid[BH3] and FHA-Bid[D59A] but not FHA-Bid into *Bid-/-* MPCs (*Bid-/-* v. *Bid-/- + FHA-Bid[BH3]*, p<0.0001 and *Bid-/-* v. *FHA-Bid[D59A]*, p=0.0008) (*Figure 6c*).

Despite decreased oxygen consumption, one possible explanation for the observed mitochondrial defects in *Bid-/-* cells could be damage from the generation of reactive oxygen species (ROS). We assessed baseline mitochondrial and cellular superoxide with MitoSOX and DHE, respectively, and found no difference between *Bid-/-* and *Bid +/+* MPCs, however mitochondrial superoxide was increased in *Bid-/-* cells under conditions of nutrient withdrawal (p<0.01) (*Figure 6—figure supplement 2a,b and c*).

Bid's phosphorylation sites S61 and S78 (Bid[AA]) have also been shown to correspond with increased ROS and respiration in hematopoietic stem cells (*Maryanovich et al., 2012*), (*Maryanovich et al., 2015*). Additionally, it was shown that truncated Bid (tBid) residues 57–73 had strong binding to MTCH2 (*Katz et al., 2012*), To determine if these phosphorylation sites are involved in full-length Bid's regulation of cristae function, we made both S61A and S78A point mutations in BH3-mutated Bid followed by stable re-introduction into *Bid-/-* MPCs (*Bid-/- + FHA-Bid[BH3AA]*). Interestingly, we find that even in the context of a BH3-mutant, these cells were highly unstable, which we attribute in part to the important role of these phosphorylation sites in the DNA damage response (*Liu et al., 2011*; *Zinkel et al., 2005*) as well as preventing cleavage of Bid and thus initiation of apoptosis (*Desagher et al., 2001*).

We measured TMRE and MitoSOX by flow cytometry, gating on cells positive for human CD25 (co-expressed with FHA-Bid). We find that compared to Bid[BH3], FHA-Bid[BH3AA] MPCs do not have altered membrane potential and show only a trend for increased ROS (p=0.1956) (*Figure 6—figure supplement 2d and e*). Thus, our results in MPCs are most consistent with a role for these phosphorylation sites in overall cell viability, by preventing caspase-8 cleavage of Bid (*Desagher et al., 2001*), rather than specifically in the regulation of mitochondrial membrane potential or ROS production.

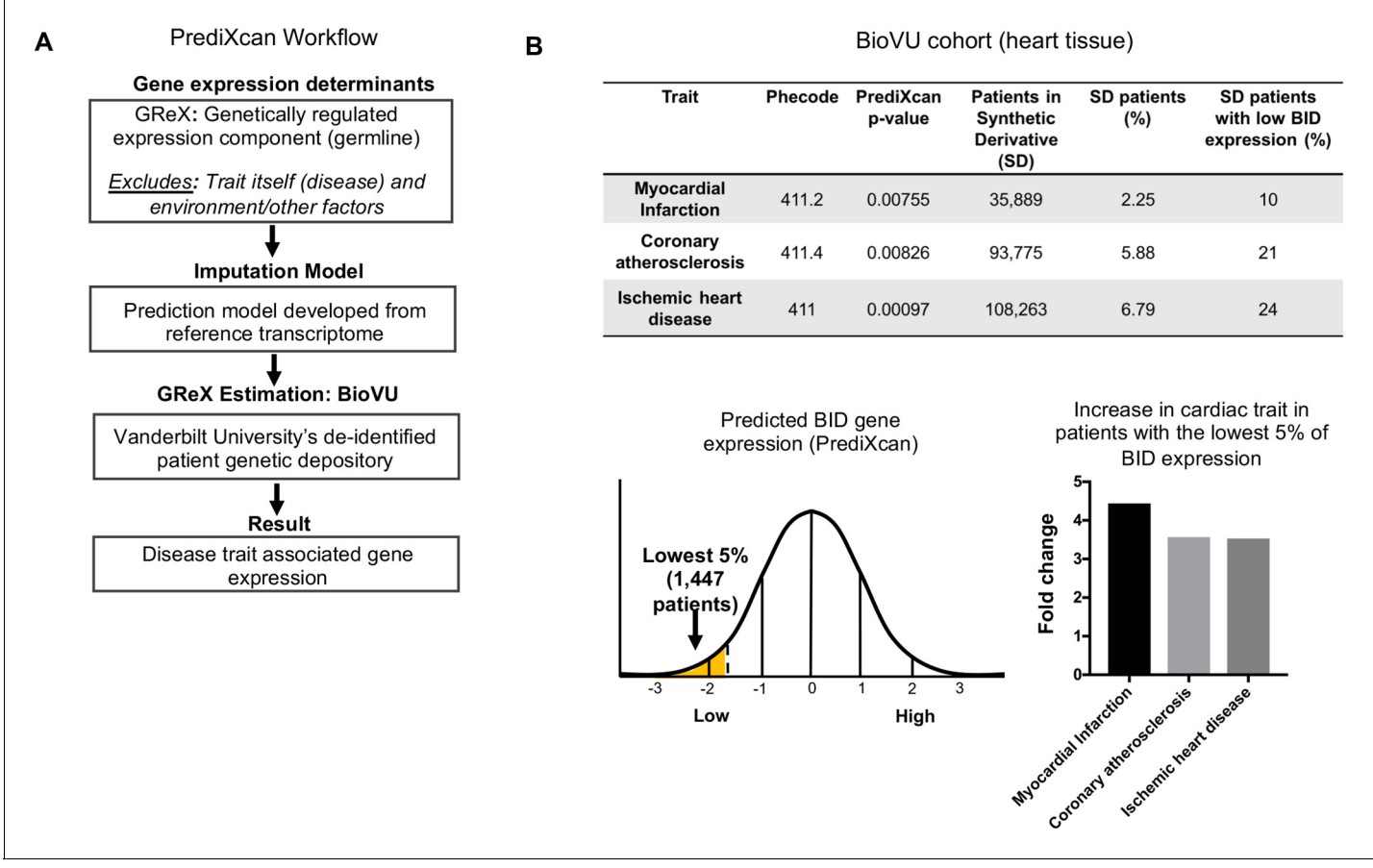

**Figure 7.** PrediXcan analysis of BID expression reveals a novel role in cardiac diseases. (**A**) Diagram of PrediXcan analysis workflow. PrediXcan estimates the genetically regulated component of gene expression (germline), excluding the impact of the disease itself and the environment on expression. (**B**) PrediXcan in a BioVU replication cohort of 29, 366 patients (in heart tissue). Patients were also binned by BID gene expression, with the lowest 5% analyzed for incidence of the cardiac traits discovered by PrediXcan. A total of 1447 patients encompassed the lowest 5% in BID expression. Myocardial infarction had the highest increased incidence, represented by graph for fold change in these patients compared to all Vanderbilt Synthetic Derivative (SD) patients (1,593,350 records). P-values were determined by logistic regression with disease status as response variable and imputed gene expression as predictor.

DOI: https://doi.org/10.7554/eLife.40907.018

The following figure supplements are available for figure 7:

**Figure supplement 1.** BID and BECN1 are significantly heritable and PrediXcan reveals significantly decreased BID expression for cardiac traits.
DOI: https://doi.org/10.7554/eLife.40907.019

**Figure supplement 2.** Secondary PrediXcan analysis of BH3-only Bcl-2 family genes shows discordant direction of effect with heart conditions.
DOI: https://doi.org/10.7554/eLife.40907.020

**Figure supplement 3.** Decreased BID expression is associated with MI in two additional independent cohorts.
DOI: https://doi.org/10.7554/eLife.40907.021

## Permeabilized cardiac fibers from *Bid-/-* mice exhibit decreased respiration and ATP production

Next, to determine whether the decreased respiration is also observed in mouse cardiac fibers, we evaluated oxygen consumption in *Bid-/-* and *Bid+/+* heart tissue. Respiration of permeabilized left ventricular (LV) cardiac fiber bundles (PmFBs) was measured in the presence of the complex I (CI) substrates malate and glutamate, as well as ADP (state 3). *Bid-/-* LV fibers also displayed significantly decreased oxygen consumption compared to *Bid+/+* LV fibers (p=0.0103) (*Figure 6d*).

To more thoroughly interrogate the mitochondrial defect from *Bid-/-* hearts, we used a customized instrument platform optimized for permeabilized muscle fibers (*Lark et al., 2016*) and simultaneously measured ATP production and $O_2$ consumption. We first analyzed PmFBs in the presence of

complex I substrates (glutamate, malate and ADP). *Bid-/-* fibers had decreased respiratory function as well as decreased ATP production (p<0.05) (*Figure 6e and f*) compared to *Bid +/+* fibers.

Rotenone, (complex I specific inhibitor) prevents electron flux through CI and we observe decreased $O_2$ consumption and ATP production as expected. Succinate directly contributes electrons to CII and was added in the presence of rotenone to interrogate CI-independent respiration. *Bid-/-* PmFBs had decreased respiration in the presence of rotenone and succinate (p<0.001), and decreased ATP production (p<0.05) (*Figure 6e and f*), consistent with our finding that irregular cristae correspond with decreased ATP synthase activity.

Oxidative phosphorylation efficiency can be defined as the ratio of ATP to O. Interestingly, despite an overall decrease in respiration and ATP production, *Bid-/-* PmFBs have similar efficiency to *Bid +/+* when using CI substrates. This is consistent with our finding that *Bid-/-* mitochondria do not have increased ROS or loss of membrane potential (*Figure 6—figure supplement 2a–d*). However, in the presence of rotenone and succinate, *Bid-/-* PmFBs have an increased ATP/O ratio (p<0.05) (*Figure 6—figure supplement 2f*). This suggests *Bid-/-* mitochondria may compensate by bypassing complex I in favor of respiratory complex I I, which is not found in respiratory supercomplexes (*Schägger and Pfeiffer, 2001*) and therefore would be less impacted by disorganized cristae.

## PrediXcan analysis reveals decreased BID expression associates with myocardial infarction

Given the observed increased fibrosis in *Bid-/-* mice, phenotypically similar to post-MI damage in humans, we investigated the clinical relevance of our findings. We applied PrediXcan (*Gamazon et al., 2015*; *Gamazon et al., 2018*) (see Materials and methods and *Figure 7a*) to test the association of genetically determined BID expression in 29,366 patients in BioVU (*Roden et al., 2008*) with MI predisposition. Because of the substantial prior support from our studies observed for Bid's role in heart function and inducing fibrotic damage with acute stress, we evaluated the association with MI risk of BID expression and used Bonferroni adjustment for the number of cardiac traits tested to assess statistical significance. Consistent with our findings in mice, we observed that decreased BID expression is significantly associated with MI (*Figure 7b*).

To quantify the extent of genetic control of BID expression, we performed SNP-based heritability analysis (*Gamazon and Park, 2016*). Genotype-Tissue Expression (GTEx) project data, despite the breadth of tissues, are still generally underpowered for this analysis (because of sample size), and we therefore utilized a larger transcriptome panel DGN (n = 922) (*Battle et al., 2014*), which is, however, available only in whole blood. The BID heritability estimate was significant ($h^2$ = 0.08 with standard error [SE] of 0.026), providing support for genetic regulation (*Figure 7—figure supplement 1a and b*).

We determined the prevalence of MI, coronary atherosclerosis and ischemic heart disease in Vanderbilt University's Synthetic Derivative (SD), which contains over 2.8 million de-identified patient records linked to electronic health records (*Roden et al., 2008*). For comparison, we evaluated the prevalence of MI, coronary atherosclerosis, and ischemic heart disease in individuals with the lowest 5% of BID expression, thus approximating the *Bid-/-* condition of our mouse model. Within this group, we find a > 4 fold increase in the prevalence of myocardial infarction compared to the rest of the Synthetic Derivative (*Figure 7b*). Decreased BID expression in heart tissue significantly associated with myocardial infarction ($p=7.55\times10^{-3}$) as well as coronary atherosclerosis ($p=8.26\times10^{-3}$), and ischemic heart disease ($p=9.7\times10^{-4}$) (*Figure 7b*). These results are notable, as they not only suggest the impact that loss of Bid would have in humans but also highlight the continuity of phenotypes observed in the *Bid-/-* mice with human patient data.

In order to more precisely characterize the effect of decreased genetically determined BID expression on cardiac phenotypes, we additionally analyzed the recently available GWAS of atrial fibrillation (N > 1 million individuals) (*Nielsen et al., 2018*). Notably, we find no significant association between BID genetically determined expression and atrial fibrillation in this dataset (p=0.63), consistent with the lack of significant association in BioVU. Thus, while we identify multiple cardiac traits associated with BID expression, BID's effect is specific to particular pathophysiologies.

To determine whether our findings are unique to BID among other BH3-only and related genes, including BECN1 (a Bcl-2-interacting protein involved in autophagy) and MTCH2 (a Bid-interacting protein) (*Grinberg et al., 2005*; *Katz et al., 2012*; *Shamas-Din et al., 2013*), we performed a secondary PrediXcan analysis. The results revealed a unique role for BID among these genes in

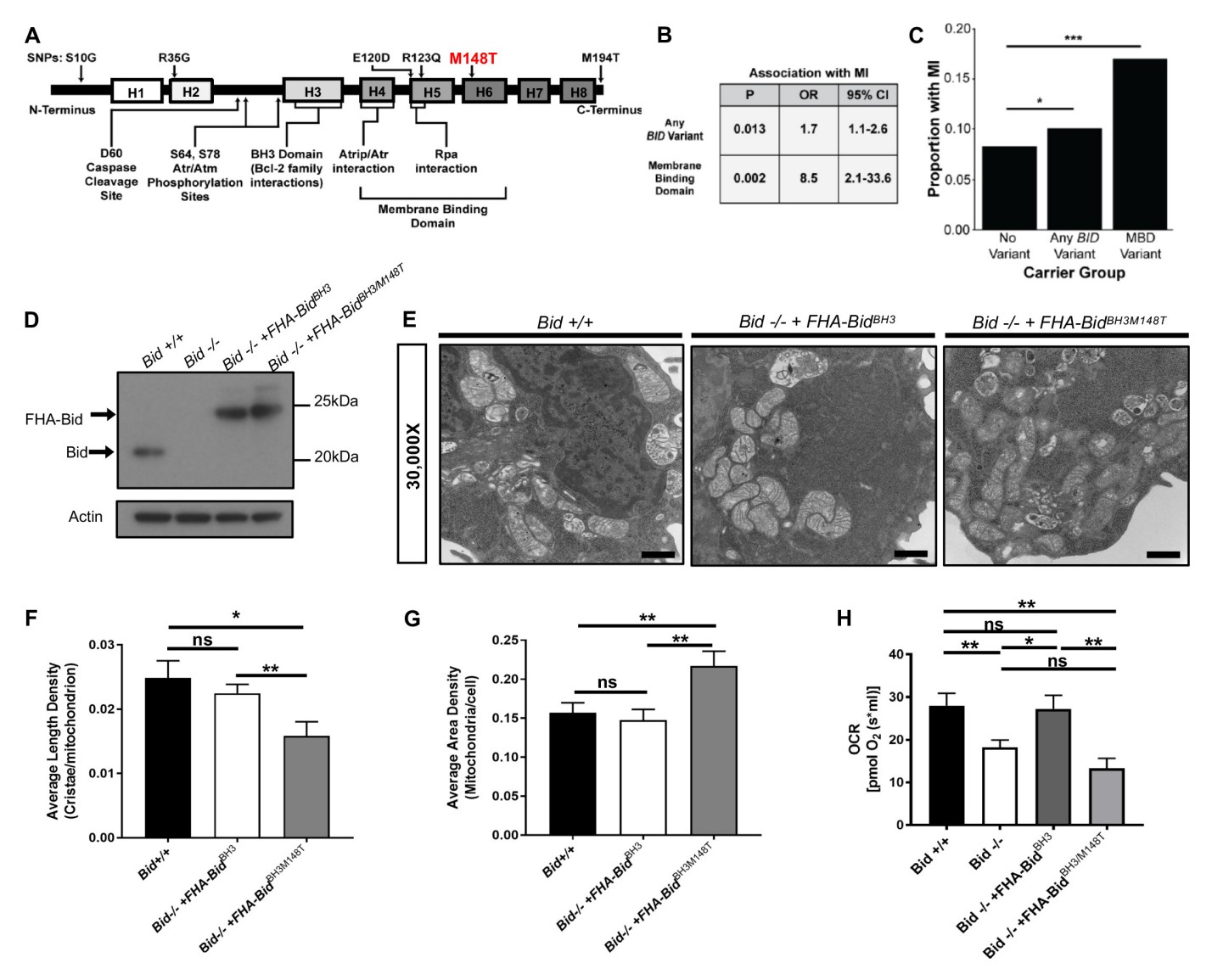

**Figure 8.** BID coding SNPs associate with myocardial Infarction (MI) in humans and reveal helix-6 SNP M148T is critical for Bid's regulation of mitochondrial function. (A) Linear representation of Bid protein structure and approximate SNP locations. Human BID SNPs and several key domains and regions of Bid are indicated. (B) Statistical values including p-value, odds ratio (OR), and 95% confidence interval (95% CI) for Bid SNP association with MI for overall carrier status of BID variants or with variants in the membrane binding domain. (C) Graphical representation of the proportion of patients with MI in carrier groups with no SNPs in BID (no variant), any BID variant, or MBD variant. (D) Western blot of expression levels of Bid for the indicated cell lines. (E) TEM of *Bid +/+*, *Bid-/- + FHABidBH^BH3*, and *Bid-/- + FHABidBH^BH3/M148T* MPCs. Representative images at 30,000X (scale bar = 500 nm). (F) Quantitation of the number of cristae per mitochondria (average length density) and (G) the mitochondrial density per cell (average area density) of the MPC lines shown in (E). (n = 40,40,15 images per cell line respectively). (H) OCR of *Bid +/+*, *Bid-/-*, *Bid-/- + FHABidBH^BH3*, and *Bid-/- + FHABidBH^BH3/M148T* MPCs for all cell lines (n = 6,12,6,5 respectively). P-values were determined by multivariable logistic regression with Bonferroni correction as described in methods for (B) and (C), one-way ANOVA with Student's t-test for (F) and G), and one-way ANOVA (p<0.05) with Student's t-test for (H). Error bars indicate ±SEM for all data. ns = not significant, *p<0.05, **p<0.01, and ***p<0.001.
DOI: https://doi.org/10.7554/eLife.40907.022

The following source data and figure supplement are available for figure 8:

**Source data 1.** Data for *Figure 8* and *Figure 8—figure supplement 1*.
DOI: https://doi.org/10.7554/eLife.40907.024
**Figure supplement 1.** Mutations in Bid corresponding to Helix-5 SNPs show less impact on respiration than Helix-6 mutation M148T.
DOI: https://doi.org/10.7554/eLife.40907.023

conferring MI risk (see Supplementary Information, Materials and methods, and *Figure 7—figure supplement 2a–f*).

## Validation in BioVU and CARDIoGRAMplusC4D GWAS

In a separate BioVU sample set (see Materials and methods and *Figure 7—figure supplement 1c and d*), we observed a significant correlation (p=0.002) between decreased genetically determined BID expression in the aorta and MI. We analyzed the publicly available CARDIoGRAMplusC4D GWAS datasets (*Schunkert et al., 2011*; *Nikpay et al., 2015*) (see Materials and methods). Consistent with the BioVU discovery and validation results, decreased genetically determined expression of BID in heart was associated (p=0.02, effect size = −0.06, SE = 0.026) with MI in CARDIoGRAMplusC4D.

Interestingly, several of the SNPs (in the locus) nominally associated with MI and CAD clustered within the adjacent BCL2L13 (Bcl2-rambo) gene, the most significant being rs2109659 (p=0.004). However, no association with MI in CARDIoGRAMplusC4D was observed with BCL2L13 (p=0.75) (*Figure 7—figure supplement 3a and b*), consistent with the SNPs being regulatory for BID.

For completeness, we report the BID associations with cardiac traits using additional tissues (*Sudlow et al., 2015*). Interestingly, all nominally significant associations with other cardiac traits in these tissues in BioVU were consistent with decreased expression of BID (*Figure 7—figure supplement 3c*).

## Bid's alpha-helix-6 is important for its ability to regulate mitochondrial function

Here we show that site-directed mutagenesis informed by exome association analysis of BID revealed that Bid's alpha-helix-6 directs its role to regulate mitochondrial function. First, we evaluated whether there was an association between coding SNPs within BID and MI risk. Using BioVU, we developed a cohort of 23,195 self-reported Caucasian subjects (median age 63 years [IQR 43 to 57 years] and 52% female) who had previously undergone genotyping (Illumina Human Exome BeadChip v1) (see Supplementary Information), of whom 1507 were MI cases. In multivariable logistic regression, a significant association was observed between carrier status (i.e. presence of any missense variant) and MI (p=0.013; OR 1.7 [95% CI 1.1–2.6]). Although this would not meet significance in an unbiased, exome-wide search, we are testing only a single gene for which we have already observed substantial evidence for its role in conferring MI risk. This gene-level association was primarily driven by variants in the membrane binding domain (MBD), including E120D, R123Q and M148T (*Figure 8a*). Carrier status for MBD variants (i.e. presence of any missense variant in the MBD) was strongly associated with MI (p=0.002; OR 8.5 [95% CI 2.1–33.6]) (*Figure 8b and c*). Notably, M148T was also associated with MI risk (p=0.029, OR = 1.47) in the recent meta-analysis of exome-chip studies involving 42,335 patients and 78,240 controls of European ancestry, consistent with the BioVU results (*Stitziel et al., 2016*).

We next evaluated whether any of these coding variants, particularly those that lie within the MBD, affect Bid's regulation of mitochondrial function. In particular, the conserved M148 residue lies within Bid's alpha-helix-6, which regulates mitochondrial association and cristae remodeling in the context of cBid during apoptosis (*Cogliati et al., 2013*; *Oh et al., 2005*; *Shamas-Din et al., 2013*).

We introduced the M148T mutation in conjunction with full-length BH3-mutated Bid, which can rescue mitochondrial function, into *Bid-/-* MPCs (*Figure 8d*). To establish that introduction of the M148T mutant does not disrupt Bid's overall structure, we evaluated apoptotic function by assessing cell death with TNF-α/Actinomycin D. As expected, *Bid-/-* MPCs were protected from death compared to *Bid +/+* MPCs (p=0.0068). Importantly, *Bid-/- + FHA-Bid^{BH3/M148T}* MPCs displayed similar death kinetics to *Bid-/- + FHA-Bid^{BH3}* MPCs which has been shown to have some sensitivity to TNF-α/Actinomycin D stimulated death (*Wang et al., 1996*). This indicates that the M148T mutation has no effect on Bid's apoptotic function in the presence of a mutated BH3 domain (*Figure 8—figure supplement 1a and b*).

We evaluated mitochondrial cristae number in *Bid +/+*, *Bid-/- + FHA-Bid^{BH3}* and *Bid-/- + FHA-Bid^{BH3M148T}* (double mutant) as in *Figure 2*, and found that the double mutant had significantly less cristae compared to *FHA-Bid^{BH3}* alone (p<0.01). Interestingly, we found that the double mutant had an increase in overall mitochondrial area density per cell, likely as a compensatory mechanism for

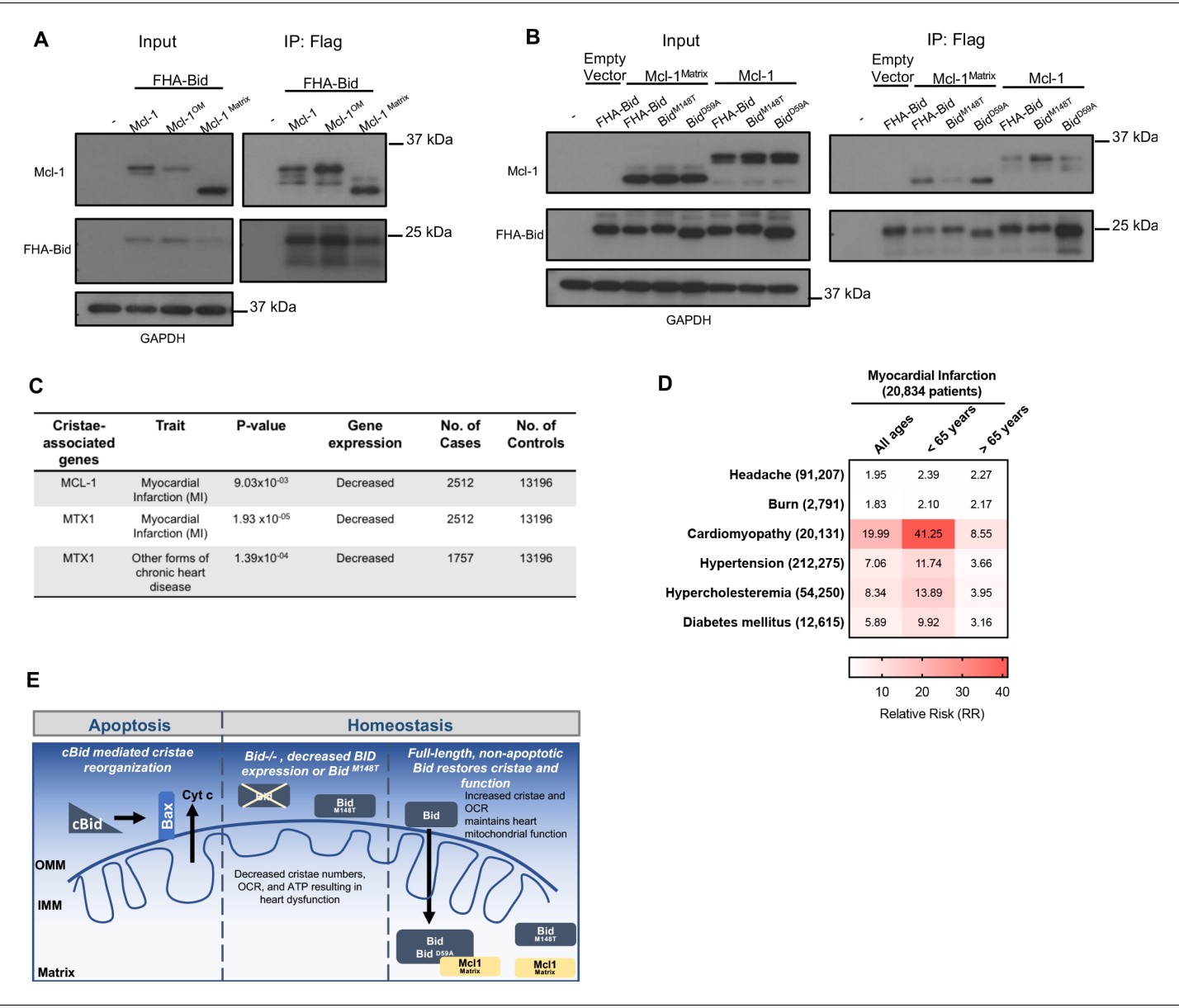

**Figure 9.** Full-length Bid interacts with Mcl-1$^{Matrix}$, which is diminished by M148T-mutated Bid. (**A**) Immunoprecipitation of FlagHA-Bid with anti-Flag M2 agarose beads from 293T whole cell lysate overexpressing FHA-Bid and one of the indicated Mcl-1 constructs: Mcl-1 (WT), Mcl-1$^{OM}$ (outer membrane), or Mcl-1$^{Matrix}$. (**B**) Immunoprecipitation as in A with the indicated Bid constructs overexpressed with either empty vector (MSCV), Mcl-1 (WT) or Mcl-1$^{Matrix}$ in 293 T cells. Input represents approximately 1/70$^{th}$ of total protein used for immunoprecipitation. (**C**) PrediXcan analysis of proteins previously found to be involved in cristae stability. MTX1 = Metaxin1. (**D**) Contingency table of patients queried in the BioVU Synthetic Derivative for MI and the indicated diseases (left) identified by ICD9 code. Patient numbers are indicated in parenthesis and values in the heat map indicate the raw relative risk (RR) values. p=3.944×10$^{-16}$ for MI v burn or headache (control diseases) and p<2.2×10$^{-16}$ for MI v all other diseases. (**E**) Proposed model for a full-length Bid's homeostatic role in regulating mitochondrial cristae structure. Bid can localize to the matrix where its association with Mcl-1 (directly or indirectly) facilitates the stabilization of respiratory complexes and cristae structure. This interaction is diminished by M148T-mutated Bid.

DOI: https://doi.org/10.7554/eLife.40907.025

The following figure supplement is available for figure 9:

**Figure supplement 1.** Table of cristae structure associated genes analyzed by PrediXcan.

DOI: https://doi.org/10.7554/eLife.40907.026

decreased cristae function (p<0.01) (*Figure 8e–g*). Respiratory efficiency of MPCs was then assessed using these mutants, directly comparing the BH3-mutant to the double mutant. Expression of the Bid^BH3/M148T^ double mutant was insufficient to restore respiratory levels comparable to *Bid +/+* or *Bid-/- + FHA-Bid^BH3^* MPCs (*Figure 8h*).

Interestingly, the two other SNPs identified in the membrane binding region of Bid also lie within a hydrophobic region of Bid as well as the region predicted to interact with MTCH2 (*Katz et al., 2012*). We made the corresponding mutations, E120D and R124Q in BH3-mutated Bid to determine if these would also result in altered mitochondrial function (*Figure 8—figure supplement 1c*). Compared to BH3-mutated Bid, Bid^BH3/E120D^ MPCs had equivalent respiration. While Bid^BH3/R124Q^ MPCs had decreased respiration (*Figure 8—figure supplement 1d*), it was not significantly different from WT MPCs. Neither Bid^BH3/E120D^ nor Bid^BH3/R124Q^ MPCs displayed altered sensitivity to TNF-α/Actinomycin D stimulated cell death (*Figure 8—figure supplement 1e and f*).

## Bid binds the matrix form of Mcl-1, which can be altered with helix-6 mutant M148T

Our observation that Bid is found within the mitoplast (*Figure 3c and d*) raised the possibility that it is interacting with mitochondrial matrix proteins known to regulate cristae structure. In particular, the anti-apoptotic Bcl-2 family member Mcl-1 has been shown to have a matrix isoform involved in respiratory chain maintenance and mitochondrial metabolism (*Escudero et al., 2018*; *Perciavalle et al., 2012*; *Thomas et al., 2013*; *Wang et al., 2013*). It is known that the BH3-domain of cBid associates with Mcl-1, to inhibit apoptosis (*Clohessy et al., 2006*).

We tested whether full-length Bid associates with WT Mcl-1, an outer mitochondrial membrane form of Mcl-1^OM^ or the matrix form of Mcl-1, Mcl-1^Matrix^. Using FlagHA-tagged Bid expressed in 293 T cells, we were able to immunoprecipitate all three forms of Mcl-1 (*Figure 9a*). This is in contrast to the other BH3-only protein Bim, which did not associate with Mcl-1^Matrix^ (*Perciavalle et al., 2012*). We then sought to determine the role of helix-6 in this association. We find that FHA-Bid^M148T^ displays decreased association with Mcl-1^Matrix^ compared to both WT-Bid and our rescue mutant, FHA-Bid^D59A^. Furthermore, FHA-Bid^M148T^ displays increased association with WT Mcl-1 relative to either WT Bid or FHA-Bid^D59A^ (*Figure 9b*). The above results are consistent with a role for helix-6 in Bid's association with Mcl-1^Matrix^, in the context of the mitochondrial cristae.

## PrediXcan reveals decreased MCL-1 gene expression is associated with myocardial infarction

Informed by our observation of Bid's interaction with mitochondrial matrix proteins known to regulate cristae structure organization, we applied PrediXcan to evaluate potential contribution to MI susceptibility for these genes (see *Figure 9—figure supplement 1*). Loss of Mcl-1 has previously been shown to result in cardiomyopathy (*Wang et al., 2013*) and impaired autophagy leading to heart failure in mice (*Thomas et al., 2013*). We find that decreased genetically determined expression of MCL-1 is significantly associated with MI (p=0.00903) (*Figure 9c*). In addition to MCL-1, we find that MTX1 (Metaxin1), a mitochondrial protein transporter that associates with the MICOS complex (*Guarani et al., 2015*), has reduced genetically determined expression significantly associated with MI (p=1.93×10^−5^).

Furthermore we utilized the Synthetic Derivative (*Roden et al., 2008*) to gain additional insights into the cardiac traits known to result from loss of Mcl-1. Using ICD9-codes, we identified 20,834 patients diagnosed with an MI (among the nearly 2.8 million patients). Using this information, we constructed a contingency table, first looking at the relative risk for control phenotypes (headache and burn) as well as known risk factors for MI including hypertension, hypercholesteremia, and diabetes mellitus (*Anand et al., 2008*). Interestingly, we find that patients with a history of cardiomyopathy have a significantly increased relative risk for MI compared to the known risk factors, connecting these two phenotypes genetically (*Figure 9d*).

Thus, we propose a model in which we have evidence from cell lines, mice, and multiple human genetics studies that converge on a role for Bid in the regulation of mitochondrial cristae structure and predisposition to MI (*Figure 9e*). We further find genetic evidence that decreased expression of two additional genes known to regulate cristae structure, MCL-1 and MTX1, is also associated with susceptibility to MI. In addition to its apoptotic function, we now add a homeostatic function for Bid

at the mitochondria which is dependent on its full-length form in the matrix, and the helix-6 residue M148, uncovered directly from human exome data. Finally, we find an association between Bid and the matrix form of Mcl-1 mediated by the helix-6 residue M148, suggesting that Bid may perform its role at the mitochondrial matrix through interaction with Mcl-1.

## Discussion

Our results add to the body of literature implicating a role for the Bcl-2 family in mitochondrial membrane remodeling in the absence of apoptosis. While full-length Bid is observed at the mitochondria homeostatically (*Maryanovich et al., 2012*; *Wang et al., 2014*; *Figure 3*), the purpose for this localization, especially given that cleaved Bid is potently apoptotic, was unclear. We find that Bid, like Bcl-X$_L$ and Mcl-1(*Chen et al., 2011*; *McNally et al., 2013*; *Perciavalle et al., 2012*), is critical for the structural and functional maintenance of mitochondrial cristae and this occurs independently of caspase-8 cleavage. The significance of this finding is strengthened by our complementary approach, which integrates cell biology with human genetics data.

In both MPCs as well as LV tissue, loss of Bid results in absent or abnormal mitochondrial cristae structure. Acute cardiac stress not only exacerbates this cristae disorganization but leads to cardiac dysfunction in *Bid-/-* mice, including increased left ventricular diameter and reduced ejection fraction. While mice are able to recover functionally, this ultimately results in increased cardiomyocyte fibrosis, damage similar to that observed after an MI. We propose that the association between Bid and MI can be linked to mitochondrial function. Real-time analysis of permeabilized cardiac fibers revealed that loss of Bid results in decreased respiration and ATP production. Thus *Bid-/-* cells and tissue function at their maximum efficiency, yet produce less energy, consistent with disrupted respiratory chain formation (*Lapuente-Brun et al., 2013*). Under conditions of stress, *Bid-/-* mitochondria are unable to meet increased energetic demand, thus decreasing the threshold to cardiac failure, and ultimately myocardial dysfunction.

To determine the human disease relevance of our findings, PrediXcan analysis (*Gamazon et al., 2015*) was applied to BioVU (*Roden et al., 2008*). The PrediXcan-derived association of BID with MI has important implications. Firstly, the association derives from common genetic variants, and therefore has potential diagnostic implications in the general population. Secondly, use of germline genetic profile to estimate BID expression removes any potential confounding effect the environment or disease itself could have on gene expression.

Importantly, we also evaluated the individuals with the lowest BID expression, thus approximating the situation in *Bid-/-* mice in humans. Strikingly, the lowest 5% of individuals had a > 4 fold increase for incidence of MI. This remarkable result further connects our genetic findings to the cardiac phenotype observed in *Bid-/-* mice.

Lastly, we sought independent validation for BID's association with MI. We used the publicly available CARDIoGRAMplusC4D GWAS datasets (*Schunkert et al., 2011*)(*Nikpay et al., 2015*), and an additional independent cohort of BioVU patients. This result was also unique to BID among other BH3-only proteins.

At the coding level, we have also identified SNPs within the membrane binding region of BID associated with MI. In particular, M148T in helix 6 was of interest as two additional downstream residues, K157 and K158, have been shown to be essential for cristae re-organization in the context of apoptosis (*Cogliati et al., 2013*). This SNP was also found to be significant in a meta-analysis of exome-chip studies of European ancestry (*Stitziel et al., 2016*). To determine the functional consequence of this SNP, we made the corresponding M148T point mutant in Bid and find it fails to fully restore cristae structure, and results in a loss of respiratory function when combined with our rescue BH3-mutant Bid. In contrast, two SNPs in the putative Bid-MTCH2 binding domain (Bid$^{E120D}$ and Bid$^{R124Q}$) did not alter mitochondrial function compared to WT MPCs.

Our results indicating the presence of Bid in the matrix prompted us to determine if the M148T mutant would also impact a possible protein-protein interaction. A strong candidate is the anti-apoptotic Bcl-2 family member Mcl-1, which was rigorously shown to have a mitochondrial matrix isoform that mediated mitochondrial cristae structure and lipid metabolism independent of Mcl-1's apoptotic function (*Escudero et al., 2018*; *Perciavalle et al., 2012*). We find that this point mutant decreases the association between Bid with Mcl-1$^{Matrix}$ compared to WT and D59A-mutated Bid (rescue mutant). Interestingly, we also observe that Bid$^{M148T}$ associates WT Mcl-1.

We propose that Bid interacts with Mcl-1 in a manner that requires not only a BH3-domain, but also helix- 6. Based on the NMR structure of Bid (*Chou et al., 1999*; *McDonnell et al., 1999*), Bid$^{M148T}$, as well as the previously implicated Bid$^{K158}$ (*Cogliati et al., 2013*), lie in approximately the same plane of helix-6, in an orientation facing away from helix-3 (BH3-domain) in solution. EPR analysis of p15 Bid also places both of these residues in the headgroup region of a lipid bilayer when cBid is inserted into a membrane (*Oh et al., 2005*). It is possible that mutating these residues decreases the affinity of Mcl-1 to full-length Bid in solution by destabilizing the hydrophobic core of Bid adjacent to helix-3. Alternatively, these mutants might also be predicted to decrease Bid's association with a membrane. This may be more critical for Bid's interaction with Mcl-1$^{Matrix}$ in regulating membranes than for its interaction on the surface of mitochondria with WT Mcl-1 and may account for the difference in affinity found by immunoprecipitation.

In sum, we have identified a homeostatic role for Bid in the regulation of mitochondrial structure and function extending initial observations in tissue culture to an in vivo model that converges on a unique role for BID in human cardiac disease. We propose that loss of Bid or decreased BID gene expression contributes to cardiac diseases, particularly MI. Furthermore, we provide evidence that this mitochondrial function for Bid is dependent at least in part upon Bid's alpha-helix-6, that mediates Bid's interaction with Mcl-1$^{Matrix}$, implicating a Bid-Mcl-1 interaction at the matrix in mitochondrial cristae organization. Finally, we find an association between decreased expression of MCL-1 and MTX-1 with susceptibility to MI, linking altered cristae structure with MI. Our integrated approach, combining multiple avenues of investigation, has identified previously unknown proteins involved in complex genetic diseases, and can be used to bridge the gap between basic biological findings and translational science.

# Materials and methods

## Key resources table

| Reagent type (species) or resource | Designation | Source or reference | Identifiers | Additional information |
|---|---|---|---|---|
| Strain, strain background (*mus musculus*, C57BL/6J) | | The Jackson Laboratory | Stock No: 000664 (Black 6) RRID:IMSR_JAX: 000664 | |
| Strain, strain background (*mus musculus*, C57BL/SJ) | *Bid-/-* | PMID: 10476969 | | |
| Cell line (mouse) | Myeloid Progenitor Cells (MPCs) | PMID: 16122425 | | |
| Gene (mouse) | BID (BH3 interacting death domain agonist) | PMID: 8918887 NCBI Reference | MGI:108093 NM_007544.4 | |
| Transfected construct | pOZ-FH-C-hCD25 | PMID: 14712665 | | Available from Addgene (plasmid #32516) |
| Transfected construct (pOZ-FH-C-hCD25 vector) | FHA-Bid | PMID: 8918887 | | |
| Transfected construct (pOZ-FH-C-hCD25 vector) | FHA-BidBH3 | PMID: 8918887 | | |
| Transfected construct (pOZ-FH-C-hCD25 vector) | FHA-BidD59A | PMID: 12519725 | | |

*Continued on next page*

*Continued*

| Reagent type (species) or resource | Designation | Source or reference | Identifiers | Additional information |
|---|---|---|---|---|
| Transfected construct (pOZ-FH-C-hCD25 vector) | FHA-BidBH3AA | This paper | | Mutant made with site directed mutagenesis of FHA-BidBH3 construct; Zinkel Laboratory; See *Table 1* for primer sequences |
| Transfected construct (pOZ-FH-C-hCD25 vector) | FHA-BidBH3/M148T | This paper | | Mutant made with site directed mutagenesis of FHA-BidBH3 construct; Zinkel Laboratory; See *Table 1* for primer sequences |
| Transfected construct (pOZ-FH-C-hCD25 vector) | FHA-BidM148T | This paper | | Mutant made with site directed mutagenesis of FHA-Bid construct; Zinkel Laboratory; See *Table 1* for primer sequences |
| Transfected construct (pOZ-FH-C-hCD25 vector) | FHA-BidBH3/E120D | This paper | | Mutant made with site directed mutagenesis of FHA-BidBH3 construct; Zinkel Laboratory; See *Table 1* for primer sequences |
| Transfected construct (pOZ-FH-C-hCD25 vector) | FHA-BidBH3/R124Q | This paper | | Mutant made with site directed mutagenesis of FHA-BidBH3 construct; Zinkel Laboratory; See *Table 1* for primer sequences |
| Antibody | Bid (goat polyclonal) | R and D systems | AF860 RRID: AB_2065622 | 1:1000 (5% milk, Western Blot (WB)) |
| Antibody | Bid (rabbit polyclonal) | PMID: 8918887 | Antibody generated by S. Korsmeyer Lab | 1:1000 (5% milk, WB) |
| Antibody | Bim (H-5, mouse monoclonal) | Santa Cruz Biotech-nology | sc-3743589 RRID: AB_10987853 | 1:100 (5% milk, WB) |
| Antibody | Bad (Clone 48, mouse) | BD Biosciences | 610391 RRID: AB_397774 | 1:500 ((5% milk, WB) |
| Antibody | Puma/bbc3, N-terminal (rabbit) | Sigma-Aldrich | P4743 RRID: AB_477351 | 1:1000 (5% milk, WB) |
| Antibody | Anti-HA (rabbit polyclonal) | Sigma-Aldrich | H6908 RRID: AB_260070 | 1:1000 (5% milk, WB) |
| Antibody | VDAC1 (rabbit polyclonal) | Abcam | ab15895 RRID: AB_2214787 | 1:1000 (5% milk, WB) |
| Antibody | β-Actin (AC-15, mouse monoclonal) | Sigma-Aldrich | A5441 RRID: AB_476744 | 1:200,000 (5% milk, WB) |
| Antibody | Anti-GAPDH (FL-335, rabbit polyclonal) | Santa Cruz Biotech-nology | sc-25778 RRID: AB_10167668 | 1:1000 (5% milk, WB) |
| Antibody | Bak, NT (rabbit polyclonal) | EMD Millipore | Cat #06–536 RRID: AB_310159 | 1:1000 (5% milk, WB) |
| Antibody | MnSOD (rabbit polyclonal) | Stressgen | ADI-SOD-111 RRID: AB_10631750 | 1:1000 (5% milk, WB) |
| Antibody | Mcl-1 (rabbit polyclonal) | Rockland Immunochemi-cals Inc | 600-401-394S RRID: AB_2266446 | 1:1000 (5% milk, WB) |
| Antibody | Opa-1 (Clone 18, mouse) | BD Biosciences | 612606 RRID: AB_399888 | 1:1000 (5% milk, WB) |

*Continued on next page*

*Continued*

| Reagent type (species) or resource | Designation | Source or reference | Identifiers | Additional information |
|---|---|---|---|---|
| Antibody | Calreticulin (D3E6, XP, rabbit monoclonal) | Cell Signaling Technology | 12238 RRID: AB_2688013 | 1:1000 (5% milk, WB) |
| Antibody | Anti-pyruvate dehydrogenase E2/E3 (mouse monoclonal) | Abcam | ab110333 RRID: AB_10862029 | 1:1000 (5% milk, WB) |
| Antibody | Amersham ECL anti-rabbit IgG, HRP-linked (from donkey) | GE Healthcare | NA934 RRID: AB_772206 | 1:10,000 (5% milk, WB) |
| Antibody | Goat anti-mouse IgG, HRP-conjugate | Novex | A16072 RRID:AB_2534745 | 1:10,000 (5% milk, WB) |
| Antibody | Donkey anti-goat IgG HRP | Santa Cruz Biotech-nology | sc-2020 RRID:AB_631728 | 1:10,000 (5% milk, WB) |
| Chemical compound, drug | Doxorubicin HCl (Dox) | APP Fresenius Kabi USA, LCC | NDC 63323-883-05 | |
| Chemical compound, drug | Epinephrine (Epi) | BPI Labs, LLC | NDC 54288-103-10 | |
| Chemical compound, drug | Fugene 6 Transfection Reagent | Promega | E2691 | |
| Chemical compound, drug | Lipofectamine 2000 Transfection Reagent | ThermoFisher Scientific | 11668027 | |
| Commercial assay or kit | QuikChange XL Site-Directed Mutagenesis Kit, 10 rxn | Agilent Technologies | 200521 | |
| Commercial assay or kit | GeneJET Plasmid Miniprep Kit | Thermo-Fisher Scientific (Thermo Scientific) | K0503 | |
| Commercial assay or kit | GenElute HP Plasmid Maxiprep Kit | Sigma-Aldrich | NA0310-1KT | |
| Commercial assay or kit | PureLink HiPure Plasmid Maxiprep Kit | Thermo-Fisher Scientific (Invitrogen) | K210006 | |
| Software, algorithm | PrediXcan | PMID: 26258848 and other | | https://github.com/hakyimlab/PrediXcan |
| Software, algorithm | S-PrediXcan | Other | | https://github.com/hakyimlab/MetaXcan |
| Other | CARDIoGRAMplusC4D | Other | | www.CARDIOGRAMPLUSC4D.ORG |
| Other | GTEx Consortium (v6p) | PMID: 29022597 and other | | http://www.gtexportal.org |

## Mice

All mice were housed, and experiments performed with approval by the IACUC of Vanderbilt University Medical Center in compliance with NIH guidelines. WT (*Bid+/+*) and *Bid-/-* mice were back-crossed onto a C57BL/6 background at least nine generations in addition to being re-derived to mice with a pure C56BL/6 background. Age and sex of mice used for experiments are indicated where applicable.

## Cell culture and Bid mutants

Hox11-immortalized MPCs were cultured in IMDM medium with 20% FBS, 100 U/ml penicillin-streptomycin, 2 mM glutamine, 0.1 mM β-mercaptoethanol, and 10% WEHI conditioned medium as a source of IL-3. Cell lines were mycoplasma tested and negative. Cell lines were also authenticated by genotyping. To generate MPCs expressing exogenous wild type or mutant Bid, Bid was cloned

**Table 1.** Site-directed mutagenesis primer sequences for *Bid*

| Primer (Bid mutant) | | Sequence |
|---|---|---|
| M148T | | Fwd: 5' GGAGAACGACAAGGCCATGCTGATAATGACAATGC 3' |
| | | Rev: 5' GCATTGTCATTATCAGCATGGCCTTGTCGTTCTCC 3' |
| E120D | | Fwd: 5' GAATGGCAGCCTGTCGGATGAAGACAAAAGGAAC 3' |
| | | Rev: 5' GTTCCTTTTGTCTTCATCCGACAGGCTGCCATTC 3' |
| R123Q | | Fwd: 5' GTCGGAGGAAGACAAAAGGAACTGCC GGCCAAAG 3' |
| | | Rev: 5' CTTTGGCCAGGCAGTTCCTTTTGTCTTCCTCCGAC 3' |
| S78A | | Fwd: 5'CCAGATTCTGAAGCTCAGGAA GAAATCATCCACAACATTGCC3' |
| | | Rev: 5'GGCAATGTTGTGGATGATTTCTTCCTGAGCTTCAGAATCTGG3' |
| S61A | | Fwd: 5'CAGACAGACGGCGCCCAGGCCAGCCGC3' |
| | | Rev: 5'GCGGCTGGCCTGGGCGCCGTCTGTCTG3' |

DOI: https://doi.org/10.7554/eLife.40907.027

into pOZ-FH-C-hCD25 using XhoI and NotI restriction sites (*Nakatani and Ogryzko, 2003*). BH3 mutant Bid has amino acids 93–96 of mouse Bid mutated from IGDE to AAAA (*Wang et al., 1996*). The D59A mutant Bid is mutated at the caspase eight cleavage site. M148T, E120D, R123Q, and BH3S61AS78A (BH3AA) were designed according to the Quickchange II Site-directed mutagenesis Kit (Agilent Technologies) using the pOZ-FH-C-Bid-BH3-mut-hCD25 as a template. Stable cell lines were generated with retroviral transduction using Fugene 6 (Promega) or Lipofectamine 2000 (Thermo Fisher Scientific). Please see *Table 1* for primer sequences.

## Cell death and proliferation assays

At the indicated times cells were, washed, incubated with Annexin V-FITC (Biovision) in 1X Annexin V staining buffer (10 mM HEPES, pH 7.4, 140 mM NaCl, 2.5 mM CaCl$_2$). Immediately prior to analysis, propodeum iodide (Sigma) was added to a final concentration of 1 μg/ml. TNF-α/Actinomycin D death assays were performed by treating cells with 25 ng/ml TNF-α and 50 ng/ml Actinomycin D in complete IMDM growth medium Samples were analyzed on a Becton-Dickinson flow cytometer and FlowJo analysis software. Cell growth was determined by trypan blue viability.

## Electron microscopy and image quantitation

Cells were washed with 0.1 M cacodylate buffer and fixed in 2.5% glutaraldehyde/0.1M cacodylate for 1 hr at room temperature and left at 4°C overnight. The samples were post-fixed in 1% osmium tetroxide and washed 3 times with 0.1 M cacodylate buffer. The samples were dehydrated through a graded ethanol series followed by incubation in 100% ethanol and propylene oxide (PO) as well as 2 exchanges of pure PO. Samples were embedded in epoxy resin and polymerized at 60°C for 48 hr.

For each sample, 70–80 nm ultra-thin sections were cut and mounted on 300-mesh copper grids. Two sections per sample were stained at room temperature with 2% uranyl acetate and lead citrate. Imaging was done on a Philips/FEI Tecnai T-12 high resolution transmission electron microscope with a side mounted 2k × 2 k AMT CCD camera. For initial cell line analysis, a total of 40 images were captured per cell type. Images were quantified at 30,000x.

LV cardiac tissue was harvested from WT or *Bid-/-* mice at 18 hours with or without Epinephrine (0.5 mg/kg) and immediately fixed and processed as described above. All images were acquired in the Vanderbilt Cell Imaging Shared Resource.

Quantitation was done with FIJI (Fiji Is just ImageJ) software using a stereology plugin (Version 0.1) to create a multipurpose stereological grid (*Gundersen and Jensen, 1987*). Horizontal grid lines were overlaid on each image using the same tile density setting for all samples. The end of each line was counted as a point and points on the grid were counted as nucleus, extracellular space, cytoplasm or mitochondria. Total reference points per image were everything except nucleus and extracellular space. Cristae were counted when intersecting the grid line or point, and each crista was counted twice to account for double membranes. Data is represented as either area density (equivalent to volume density), which is the number of mitochondria divided by the number of reference points. Length density (which is equivalent to surface density) was calculated as two times the

number of cristae intersections divided by the total length of line for all possible intersections. For LV tissue cristae quantitation, 150 individual cristae were measured per treatment condition using the measurement tool in ImageJ software.

## Western Blot and Co-immunoprecipitation

MPCs were treated as indicated and clarified cell extracts were prepared by lysis in RIPA buffer supplemented with protease (Complete Mini, Roche) and phosphatase (PhosSTOP, Roche) inhibitor followed by centrifugation at 12,000 rcf. Heart tissue extract was also prepared in the same way. Proteins were resolved by SDS-PAGE and transferred to PVDF membrane. Immunoblots were probed with the indicated antibodies and developed using chemiluminescent HRP substrate and autoradiography film. Co-immunoprecipitation was performed on 293T cells transfected by Fugene 6 (Promega) with the following Bid constructs: FlagHA-Bid, FHA-Bid$^{D59A}$, FHA-Bid$^{M148T}$ and the following Mcl-1 constructs (a kind gift from Dr. Joseph Opferman): pMSVC-puro (empty vector), Mcl-1 (WT), Mcl-1$^{OM}$, Mcl-1$^{Matrix}$. Input was removed from equal concentrations of whole cell lysate, followed by immunoprecipitation with Flag-M2 agarose beads (Sigma).

Antibodies used: anti-Bid goat (R and D Systems) or anti-Bid rabbit polyclonal antibody (*Wang et al., 1996*), anti-Bim H-5 (Santa Cruz), anti-Bad Clone 48 (BD Biosciences), anti-Puma/bbc3 (Sigma), N-terminal (Sigma), anti-HA tag (Sigma), anti-VDAC1 (Abcam), anti-β-Actin (Sigma), anti-GAPDH (Santa Cruz), anti-Bak, NT (EMD Millipore), anti-MnSOD (Stressgen), anti-Mcl-1 (Rockland Immunochemicals Inc) anti-Opa1 (BD Biosciences), anti-Calreticulin (Cell Signaling Technology), anti-PDH E2/E3 (Abcam), anti-HRP conjugated anti-rabbit (GE Healthcare), and HRP conjugated anti-mouse (Novex), and HRP conjugated anti-goat (Santa Cruz).

## Mitochondrial isolation

Mitochondria were isolated by differential centrifugation from both tissue and cell lines. Unless stated otherwise, all isolations were done at 4°C. Mouse liver mitochondria were isolated using a protocol adapted from Brookes et al. (*Brookes et al., 2002*)and heart mitochondria were isolated based on a protocol by JW Palmer et al. (*Palmer, 1977*).

Liver tissue: harvested livers were placed in ice cold isolation buffer (IB) (200 mM sucrose, 5 mM HEPES-KOH, pH 7.4, and 1 mM EGTA) and homogenized in a glass-glass dounce homogenizer. The homogenized tissue was centrifuged at 1,000 g and clarified supernatant was centrifuged at 10,000 g to pellet mitochondria followed by two wash spins in IB at 10,000 g to obtain a final crude mitochondrial pellet. Light membrane was removed based on the protocol by Wieckowski et al. (*Wieckowski et al., 2009*). After a crude pellet was obtained, mitochondria were resuspended in MRB buffer (250 MM mannitol. 5 mM HEPES (pH 7.4) and 0.5 mM EGTA) and further purified in 30% Percoll gradient (vol/vol), spun for 30 min at 95,000 g. Purified mitochondria were isolated with a Pasteur pipette from the bottom of the tube followed by two wash spins at 6,3000 g for 10 min. Mitochondria were resuspended in MRB buffer and stored at −80°C.

Heart tissue: Hearts were dissected, rinsed with buffer A (220 mM mannitol, 70 mM sucrose, 5 mM MOPS, 2 mM EGTA and 0.1% BSA, pH 7.4) and minced into small pieces. Tissue was homogenized in a glass-glass dounce homogenizer. Tissue was then centrifuged at 500 g, supernatant was retained, and the pellet was washed and repeated. Supernatant from both spins were combined at 3,000 g to obtain a final mitochondrial pellet.

MPCs: MPC mitochondria were prepared based upon the protocol by Wieckowski et al. (*Wieckowski et al., 2009*). At least $2 \times 10^8$ cells were harvested, rinsed in cold PBS buffer and resuspended in isolation buffer (225 mM mannitol,75 mM sucrose, 0.1 mM EGTA, and 30 mM Tris-HCl, pH 7.4) containing 20 µg/ml digitonin to permeabilize the outer membrane. After a 25 min incubation on ice, cells were homogenized with a glass-glass dounce homogenizer until >90% of the cells were damaged (determined by trypan blue visualization). Cell debris was removed with two 5 min spins at 600 g followed by a 7,000 g spin for 10 min. The mitochondrial containing pellet was washed in buffer, and spun at 7,000 g, washed and followed by a final spin at 10,000 g. The mitochondrial pellet was stored in MRB buffer at −80°C. Cytosolic proteins were retained from the supernatant after debris removing spins and spun at 100,000 g for 1 hr to separate light membranes. Protein concentration for all isolations was determined by Bradford reagent.

### Proteinase K treatment

Mitochondria were isolated from WT mouse liver, with fragmented mitochondria removed from the pellet after the first fast spin. Isolated mitochondria (4 mg/ml) were then treated with 0.5 mg/ml Proteinase K (Macherey-Nagel GmbH and Co. KG) in the presence or absence of 1% SDS for 20 min on ice followed by quenching with 5 mM PMSF.

### Mitochondrial subfractionation

Crude liver mitochondria were isolated from two WT mice, and fragmented mitochondria were removed from the pellets after the first fast spin. Pellets were combined, and treated as described in *Perciavalle et al. (2012)* with the following modification: 1. The mitoplast fraction was washed 2x in buffer (10 mM $KH_2PO_4$ with one-third volume of 10 mM $MgCl_2$) to remove contaminating OMM and 2. isolated OMM was resuspended in buffer followed by a second spin at 100,000 g for 1 hr for further purification.

### Echocardiography and Epinephrine treatment

Echocardiograms on male *Bid +/+* (WT) and *Bid-/-* C57BL/6 mice were performed under 2–3% isoflurane anesthesia using an a VisualSonics Vevo 770 instrument housed and maintained in the Vanderbilt University Institute of Imaging Science core lab. Measurements of the left ventricular internal diameter end diastole (LVIDd) and the left ventricular internal diameter end systole (LVIDs) were determined from M-mode tracings in triplicate for each mouse. Mice were echoed before (baseline), and 18 hr (stress condition) and 5 days after (recovery) an IP injection of Epinephrine at 0.5 mg/kg per mouse.

### Doxorubicin treatment

Female WT (*Bid+/+*) and *Bid-/-* C57BL/6 approximately 12–18 weeks of age were treated with 3 doses of Doxorubicin at 7.5 mg/kg 5 days apart. Echocardiogram was performed three days after the final dose. Echocardiography was performed using the VEVO2100 digital ultrasound system (Visual Sonics; Toronto, Ontario). Studies were performed using the MS400 18–38 MHz transducer. M-mode images were then processed using the Visual Sonics Software ver2.2. All measurements were made in at least duplicate using the LV trace function.

### Histology and fibrosis quantitation

Hearts were excised from mice, weighed, and fixed for a minimum of 12 hr (overnight) in 10% formalin and embedded in paraffin. Coronal sections of hearts were cut and stained using H and E and Masson trichrome blue stain by the Translational Pathology Shared Resource (TPSR) at Vanderbilt University. Trichrome stained slides were scanned at 40X magnification using the Aperio CS2 Brightfield Scanner or whole slide imaging was performed in the Digital Histology Shared Resource at Vanderbilt University Medical Center (www.mc.vanderbilt.edu/dhsr). Representative 2x and 60x H and E and Trichrome images were acquired on an Olympus BX43 brightfield microscope with a Spot Insight camera.

Aperio Imagescope software version 12.3.28013 was used to define regions of within the left ventricle, excluding edges where stain uptake may have been falsely increased or any visible artifact within the section. Slides were run through a positive pixel algorithm and input parameters were adjusted to detect aniline blue staining and positive pixels are counted and grouped as weak, medium, or strong intensity. Positivity is defined as the total number of positive pixels divided by the total number of pixels in the region of interest.

### Multidimensional protein identification technology (MudPIT)

Equal amounts of *Bid +/+* and *Bid-/-* MPC mitochondrial protein were isolated and frozen. Samples were trypsinized and analyzed by MudPIT. MS/MS spectra were identified using SEQUEST software which queried a Uniprot-mouse-reference-canonical_20121112_rev database (Unknown version, 86222 entries). Results were visualized in Scaffold 4.5.1 software (Proteome Software Inc.) and protein identification was limited to two unique peptides per protein and a 5% FDR (false discovery rate) for both peptides and proteins. For analysis, samples were ranked based upon Fisher's exact test done in Scaffold with a significance value of $p < 0.05$. The Mouse MitoCarta 2.0 (*Calvo et al.,*

2016; Pagliarini et al., 2008) (Broad Institute) was used to verify genes encoding mitochondrial proteins.

## Native-PAGE and In-gel activity assay

Mitochondria from heart and liver tissue were prepared as described. Complexes were extracted based on the protocol by Wittig, et al. (Wittig et al., 2006)and run on native gels followed by incubation with complex specific substrates.

Specifically, mitochondrial protein was extracted with digitonin at a 6.0 g/g detergent/protein ratio for complexes I and IV extraction and 2.5 g/g ratio for complex V. After solubilization, samples were spun at 20,000 g for 20 min. Supernatant was retained and protein concentration was determined by Bradford reagent (BioRAD). Samples were supplemented with 50% glycerol and 5% Coomassie blue G-250 dye. Equivalent protein concentrations were then loaded onto pre-cast NativePAGE 4–16% Bis-Tris gel (Invitrogen by ThermoFisher Scientific). Samples were run at 4°C for 30 min at 100V and 2 1/2 hr at 300V. Cathode and anode buffers per Wittig, et al. For Coomassie band visualization, gels were stained with NOVEX Colloidal blue staining kit (Invitrogen) and destained overnight. All in gel activity assays were performed at room temperature. Complex I was developed in 5 mM Tris-HCl buffer (pH 7.4) supplemented with 10 mg/ml NADH (Roche Diagnostics) and 25 mg of Nitro Blue Tetrazolium (Sigma) for 10 min. Complex IV was developed in 50 mM Sodium Phosphate buffer (pH 7.2) supplemented with 5 mg of Diaminobenzidine (DAB) (Sigma) and 100 µl of horse heart cytochrome c (Sigma) for at least 30 min. Complex V activity was determined by equilibration of native gels in a 35 mM Tris/270 mM Glycine buffer (RPI) (pH 8.3) for 1 hr followed by addition of 14 mM $MgSO_4$ (Fisher), $Pb(NO_3)_2$ (Sigma) and 8 mM ATP (Roche), adjusted to pH 8.6 and incubated until a precipitate appeared. Reactions were quenched with fixation in methanol and gels were scanned for quantitation. Quantitation was done on replicate samples (n = 3 WT and Bid-/- mice for CI and CIV, n = 4 WT and Bid-/- for CV) run on the same gel for accuracy, however experiments were done a minimum of three independent times. Analysis was done using the gel tool function of ImageJ software and graphs were generated with GraphPad Prism.

## High resolution respirometry

MPCs: To determine the basal respiration rate of MPCs, oxygen consumption rates (OCR) were measured in an Oroboros O2K oxygraph (Oroboros Instruments). For each genotype, $2 \times 10^6$ viable cells, determined by trypan blue exclusion, were added to oxygraph chambers containing 2 ml of culture medium. The average OCR was measured over an interval of stable oxygen flux following addition of cells to the chamber.

Cardiac fibers: Initial basal respiration of cardiac myocytes was performed on 2–3 mg of heart fibers extracted from the left ventricle of Bid +/+ and Bid-/- mice (Veksler et al., 1987). Fibers were prepared in ice-cold relaxation and preservation solution (2.77 mM $CaK_2EGTA$, 7.23 mM $K_2EGTA$, 6.56 mM $MgCl_2$, 5.7 mM $Na_2ATP$, 14.3 mM phosphocreatine, 20 mM taurine, 0.5 mM dithiothreitol, 50 mM K-MES and 20 mM imidazole, pH 7.1). Fibers were permeabilized by incubation at 4°C for 20 min in relaxation and preservation solution containing 50 µg/ml saponin. Fibers were transferred into mitochondrial respiratory solution (MiRO5: 0.5 mM EGTA, 3 mM $MgCl_2$, 60 mM K-lactobionate, 20 mM taurine, 10 mM $KH_2PO_4$, 20 mM HEPES, 110 mM Sucrose, and 1 g/L BSA, adjusted to pH 7.1 with KOH) Oroboros O2K oxygraph chambers containing MiRO5 buffer were supplemented with 10 mM glutamate, 4 mM malate, and 2 mM ADP. Respiration rate was determined during stabilized oxygen flux.

ATP/O of cardiac fibers: Mitochondrial ATP production and $O_2$ consumption were determined as described previously (Lark et al., 2016). Briefly, the left ventricle was excised and placed in ice-cold Buffer X containing (in mM): 7.23 $K_2EGTA$, 2.77 $CaK_2EGTA$, 20 Imidazole, 20 Taurine, 5.7 ATP, 14.3 Phosphocreatine, 6.56 $MgCl_2$-$6H_2O$ and 50 MES (pH 7.1, 295 mOsm). Under a dissecting microscope, valves and wall muscle were removed and remaining muscle separated into small bundles and weighed. Less than 3 mg wet weight of tissue was used for each experiment. Fiber bundles were incubated in Buffer X supplemented with 40 µg/ml saponin for 30 min. PmFBs were then washed in ice-cold Buffer Z containing (in mM): 110 K-MES, 35 KCl, 1 EGTA, 5 $K_2HPO_4$, 3 $MgCl_2$-$6H_2O$, and 5 mg/ml Bovine serum albumin (BSA, pH 7.4, 295 mOsm) and remained in Buffer Z on a rotator at 4°C until analysis (<4 hr).

$O_2$-equilibrated Buffer Z was supplemented with: 5 U/ml HK, 5 U/ml G6PDH, 5 mM D-Glucose, 2 mM $NADP^+$ and 20 mM Creatine Monohydrate. This buffer permitted coupling of glucose-dependent, HK-catalyzed ATP hydrolysis to G6PDH-catalyzed reduction of $NADP^+$ to NADPH in a 1:1 stoichiometry. To measure ATP synthesis, auto-fluorescence of NADPH (340/460 ex/em) was measured continuously at 30°C simultaneously with $O_2$ consumption using a customized system integrating monochromatic fluorescence (FluoroMax-4, Horiba Jobin Yvon, Edison, NJ) via a fiber optic cable (Fiberguide Industries) with high-resolution respirometry (Oroboros Oxygraph-2k, Innsbruck, Austria) (*Figure 1B*). Complex I-supported respiration was established with glutamate (10 mM) and malate (4 mM). ADP (75 µM) was added to determine Complex I-supported ATP synthesis and $O_2$ consumption. Rotenone (0.5 µM) was added to inhibit Complex I, followed by the addition of succinate (10 mM) to assess Complex II-supported respiration. Rates of ATP synthesis (JATP) were quantified by applying a standard curve generated from ATP titrations in the presence of the enzyme-coupled system and the respiratory substrates.

For each step of the experimental protocol, $JO_2$ or JATP were obtained from identical time points and are reported as the mean of >20 s of steady-state data (>10 individual data points). Instrumental background rates (prior to any substrate additions) were subtracted from all subsequent values for $JO_2$ and JATP and data were normalized to PmFB weight. ATP:O ratio was calculated by dividing the rate of ATP synthesis by the rate of atomic oxygen consumed using the formula: ATP/O = $JATP/(JO_2*2)$

## Measurement of cellular ROS and membrane potential

Intracellular ROS was determined by staining MPCs with either 2 µM MitoSOX or 5 µM DHE for 30 min followed by washing and analysis by flow cytometry. As a positive control, cells were also treated with Antimycin A for 1.5 hr prior to staining with MitoSOX. Membrane potential was measured by staining for 30 min with 50 nM TMRE. For *Bid-/- + FHA-Bid*$^{BH3}$ and *Bid-/- + FHA-Bid*$^{BH3AA}$ MPCs. Cells were also stained for human CD25 and measurements were obtained from CD25 +cells. All samples were analyzed on a Becton-Dickinson flow cytometer and FlowJo analysis software.

## PrediXcan

We performed PrediXcan analysis (*Gamazon et al., 2015*) to evaluate potential roles for Bid in myocardial infarction. PrediXcan proposes gene expression as a mechanism underlying disease risk by testing the genetically determined component of expression for association with disease risk. An observed association implies a likely causal direction of effect from the gene expression trait to disease risk since, as can be reasonably assumed, disease risk does not alter germline genetic profile. The genetic component of BID expression was estimated from an imputation model (Elastic Net (*Gamazon et al., 2015*) with mixing parameter α = 0.5) trained on a reference transcriptome data set (the Genotype-Tissue Expression (GTEx) project (*Gamazon et al., 2018*; *Gamazon et al., 2015*). Imputation performance for each analyzed tissue was evaluated using 10-fold cross-validation (between imputed expression and directly measured expression), as previously described (*Gamazon et al., 2015*).

The imputation model derived from GTEx left heart ventricle was then applied to genome-wide association study data from BioVU, a Vanderbilt University resource that links human DNA samples and genetic data to de-identified electronic health records (EHRs). The development of BioVU has been previously described (*Roden et al., 2008*). We applied PrediXcan on 29,366 individuals (of whom 5146 are MI cases) to impute the genetically determined BID expression and to test for correlation with the phenotype of interest. We performed logistic regression with disease status as the response variable and the inferred genetic component of gene expression as predictor. We also evaluated the patients with the lowest BID expression, that is, in the bottom 5% of the expression distribution and closest to a BID 'knockout', to test for enrichment of MI and to directly validate the observed increased fibrotic damage akin to human MI observed in *Bid-/-* mice. To determine whether the observed association implicated a specific pathophysiology, we applied summary-statistics based PrediXcan (*Barbeira et al., 2018*) with atrial fibrillation using a recently released GWAS data in more than 1 million patients (*Nielsen et al., 2018*).

In a secondary analysis, we also tested the other members of the BH3-only Bcl-2 family as well as the BID-interacting protein MTCH2. The connection of MTCH2 with obesity has been explored in

the literature (*Bauer et al., 2009*), prompting us to evaluate the PrediXcan association with BMI using the GIANT Consortium dataset (*Locke et al., 2015*).

## Comorbidity analysis in the Synthetic Derivative

The Synthetic Derivate consists of approximately 2.8 million de-identified records that contain basic clinical and demographic information of individuals seen at Vanderbilt University Medical Center. This resource was used to determine the number of patients with the following ICD-9 codes as well as their basic demographic information (age, sex, and ethnicity): Burn (949), Headache (784), Myocardial Infarction (410), Cardiomyopathy (425), Hypertension (401.9), Diabetes mellitus (250), Hypercholesteremia (272). Caucasian patient numbers were then used to create a 2 × 2 contingency table binned by age group to determine the relative risk (RR) of each ICD-9 code with MI. Raw RR risk scores and patient numbers are as indicated in the figure.

## Replication of gene-level association and search for cardiac phenotype associated regulatory variation

The CARDIoGRAMplusC4D Consortium consists of multiple large-scale genetic association studies (e.g., 14 CAD GWAS studies) of individuals of European descent totaling 22,233 cases and 64,762 controls and a later (larger but more heterogeneous) meta-analysis of GWAS studies of European, South Asian, and East Asian decent totaling 60,801 cases and 123,504 controls. These data provide a resource to identify new SNP associations with coronary artery disease or myocardial infarction and facilitate replication of the gene-level (PrediXcan) association (*Nikpay et al., 2015*; *Schunkert et al., 2011*).

## BioVU BID coding SNP analysis

The human clinical cohort was derived from BioVU. Genotyping was performed with the Illumina Human Exome BeadChip v1 by the Vanderbilt DNA resources core (VANTAGE) using standard quality control procedures.

Pre-specified clinical syndromes of cardiac injury were heart failure and MI. Phenotypes were defined by extraction of International Classification of Disease (ICD9) billing codes and application of a code translation table used for phenome-wide association scanning (Pews), a validated method of mapping ICD9 codes to clinical phenotypes within the EMR environment (*Denny et al., 2013*; *Denny et al., 2010*),(*Wei et al., 2017*).

Analyses of genotype-phenotype associations from the coding SNPs were performed using the R statistical package. Due to the individual rarity of variants, SNPs were collapsed prior to association testing. Pre-specified SNP groupings were: (1) presence of one or more of any genotyped missense variants in the *BID* gene, and (2) presence of one or more genotyped SNPs in the MBD. Association testing between SNPs and clinical phenotypes was performed using multivariable logistic regression with age, gender, systolic blood pressure, cholesterol levels, body mass index (BMI), and hemoglobin A1C included as covariates (in the case of heart failure, prior MI was also included as a covariate). A Bonferroni correction was applied to account for multiple testing, resulting in an adjusted p-value for significance of 0.0125.

We also utilized the recent meta-analysis of exome-chip studies of MI, involving 42,335 cases and 78,240 controls to replicate the coding SNP associations (*Stitziel et al., 2016*).

## Additional statistical methods

Within each experiment, all pairwise comparisons were made by the indicated statistical test and all relevant and significant comparisons are indicated on the figures or in figure legends. All biological replicates (denoted as n) are defined as the same experimental method independently tested on different samples of the same type of cell or mouse model. It should also be noted that one *Bid-/-* mouse was not included in the statistical analysis of echocardiogram data (*Figure 4*) at 18 hr as it was a statistical outlier (Grubbs' outlier test, p<0.05).

Graphs and statistical analysis were completed using GraphPad Prism software and the following denote statistical significance: ns = not significant, *p<0.05, **p<0.01, ***p<0.005, ****p<0.001. All error bars indicate SEM (standard error of the mean).

## Study approval

Human blood and tissue samples for BioVU were obtained with written informed consent under protocols approved by the Vanderbilt University Medical Center IRB, and PrediXcan analysis for BioVU is encompassed in VUMC IRB# 151187. As indicated in the IRB, this study does not meet the definition of human subject's research.

The Vanderbilt University Institutional Animal Care and Use Committee approved all experiments (IACUC #M16000037, M/14/231, V/17/001, M1600220).

## Data availability

The authors declare that all relevant data are available within the article and its supplementary information files.

Publicly available data on coronary artery disease/myocardial infarction have been contributed by CARDIoGRAMplusC4D investigators and have been downloaded from www.CARDIOGRAMPLUSC4D.ORG.

GTEx Consortium (v6p) transcriptome/genotype data is available through the GTEx portal (htt://www.gtexportal.org) and through dpGap (*Gamazon et al., 2018*).

Model definition files are described in *Gamazon et al., 2015*.

Code for the following analyses is publicly available:

PrediXcan: https://github.com/hakyimlab/PrediXcan

S-PrediXcan: https://github.com/hakyimlab/MetaXcan

# Acknowledgements

Transmission electron microscopy experiments were performed in part through the use of the VU Cell Imaging Shared Resource (supported by NIH grants CA68485, DK20593, DK58404, DK59637 and EY08126). Heart histology was performed through the Vanderbilt Translational Pathology Shared Resource supported by NCI/NIH Cancer Center Support Grant 2P30 CA068485-14 and the Vanderbilt Mouse Metabolic Phenotyping Center Grant 5U24DK059637-13. We thank Dr. Jennifer Pietenpol and Dr. James West for providing helpful feedback and resources. We thank Dr. James Atkinson and Dr. Kelli Boyd for providing histological analysis of mouse heart sections, Dr. Janice Williams for assistance with electron microscopy experiments, and Jaketa French for assistance in collecting echocardiography data. We thank Dr. Deborah Murdock and Dr. Prasanth Potluri for their helpful discussions. We also thank Dr. Joe Opferman for providing us with Mcl-1 constructs and technical support. E.R.G. benefited from a fellowship at Clare Hall, University of Cambridge. This article was prepared while Josh Fessel was employed at Vanderbilt University Medical Center. The opinions expressed in this article are the author's own and do not reflect the view of the National Institutes of Health, the Department of Health and Human Services, or the United States government.For the exome chip NEJM data: Data on coronary artery disease/myocardial infarction have been contributed by the Myocardial Infarction Genetics and CARDIoGRAM Exome investigators and have been downloaded from www.CARDIOGRAMPLUSC4D.ORG. For GWAS data on MI: Data on coronary artery disease/myocardial infarction have been contributed by CARDIoGRAMplusC4D investigators and have been downloaded from www.CARDIOGRAMPLUSC4D.ORG For GTEx data: The Genotype-Tissue Expression (GTEx) Project was supported by the Common Fund of the Office of the Director of the National Institutes of Health, and by NCI, NHGRI, NHLBI, NIDA, NIMH, and NINDS. The data used for the analyses described in this manuscript was release v6p. BioVU: Vanderbilt University Medical Center's BioVU projects are supported by numerous sources (https://victr.vanderbilt.edu/pub/biovu/?sid=229)

## Additional information

### Funding

| Funder | Grant reference number | Author |
|---|---|---|
| National Heart, Lung, and Blood Institute | 1R01HL088347 | Sandra S Zinkel |

| | | |
|---|---|---|
| U.S. Department of Veterans Affairs | 1I01BX002250 | Sandra S Zinkel |
| National Institute of General Medical Sciences | 2P01 GM015431 | L Jackson Roberts II II |
| National Institute of Mental Health | R01 MH101820 | Eric Gamazon |
| American Heart Association | 16POST299100001 | Daniel S Lark |
| Francis Family Foundation | | Josh Fessel |
| National Institute of Diabetes and Digestive and Kidney Diseases | GRU2558 | Daniel S Lark |
| National Heart, Lung, and Blood Institute | K08HL121174 | Josh Fessel |
| National Heart, Lung, and Blood Institute | 1 R01HL133559 | Sandra S Zinkel |
| National Institute of Mental Health | R01 MH090937 | Eric Gamazon |

The funders had no role in study design, data collection and interpretation, or the decision to submit the work for publication.

## Author contributions

Christi T Salisbury-Ruf, Conceptualization, Data curation, Formal analysis, Supervision, Validation, Investigation, Visualization, Methodology, Writing—original draft, Writing—review and editing; Clinton C Bertram, Conceptualization, Formal analysis, Validation, Investigation, Writing—original draft, Writing—review and editing; Aurelia Vergeade, Qiong Shi, Marlene L Heberling, Investigation, Writing—review and editing; Daniel S Lark, Conceptualization, Resources, Data curation, Formal analysis, Funding acquisition, Validation, Investigation, Visualization, Methodology, Writing—review and editing; Niki L Fortune, Conceptualization, Formal analysis, Validation, Investigation, Visualization, Methodology, Writing—review and editing; G Donald Okoye, Formal analysis, Investigation, Writing—review and editing; W Gray Jerome, Conceptualization, Data curation, Formal analysis, Supervision, Validation, Investigation, Visualization, Methodology, Writing—review and editing; Quinn S Wells, Conceptualization, Data curation, Formal analysis, Validation, Investigation, Methodology, Writing—review and editing; Josh Fessel, Conceptualization, Resources, Data curation, Formal analysis, Supervision, Funding acquisition, Validation, Investigation, Methodology, Writing—review and editing; Javid Moslehi, Formal analysis, Supervision, Validation, Investigation, Methodology, Writing—review and editing; Heidi Chen, Formal analysis, Writing—review and editing; L Jackson Roberts II, Conceptualization, Resources, Formal analysis, Supervision, Funding acquisition, Investigation, Methodology, Project administration, Writing—review and editing; Olivier Boutaud, Conceptualization, Resources, Formal analysis, Supervision, Investigation, Visualization, Project administration, Writing—review and editing; Eric R Gamazon, Conceptualization, Resources, Data curation, Software, Formal analysis, Supervision, Funding acquisition, Validation, Investigation, Visualization, Methodology, Writing—original draft, Project administration, Writing—review and editing; Sandra S Zinkel, Conceptualization, Resources, Data curation, Formal analysis, Supervision, Funding acquisition, Validation, Investigation, Visualization, Methodology, Writing—original draft, Project administration, Writing—review and editing

## Author ORCIDs

G Donald Okoye (iD) http://orcid.org/0000-0003-1078-688X
Sandra S Zinkel (iD) http://orcid.org/0000-0002-2818-9795

## Ethics

Animal experimentation: All mice were housed and experiments performed with approval by the IACUC Protocol # M1600037, M1600220, M/14/231, and # V-17-001 of Vanderbilt University Medical Center and the Tennessee Valley VA in compliance with NIH guidelines.

## Decision letter and Author response

Decision letter https://doi.org/10.7554/eLife.40907.042
Author response https://doi.org/10.7554/eLife.40907.043

## Additional files

### Supplementary files

• Transparent reporting form
DOI: https://doi.org/10.7554/eLife.40907.028

### Data availability

The authors declare that all relevant data are available within the article and its supplementary information files. Publicly available data on coronary artery disease / myocardial infarction have been contributed by CARDIoGRAMplusC4D investigators and have been downloaded from www.CARDIOGRAMPLUSC4D.ORG. GTEx Consortium (v6p) transcriptome/genotype data is available through the GTEx portal (htt://www.gtexportal.org) and through dpGap (GTEx Consortium, Nature 2017). Due to the GTEx Consortium's donor consent agreement, the raw data and attributes which may be used to identify the participants are not publicly available. Requests for access can be made through the dbGaP: https://www.ncbi.nlm.nih.gov/projects/gap/cgi-bin/study.cgi?study_id=phs000424.v6.p1 and are assessed bu a Data Access Committee (National Human Genome Research Institute; nhgridac@mail.nih.gov). The summary statistics results for eQTL data (v6p) are available through the GTEx portal: https://gtexportal.org/home/datasets. Investigators may obtain access to UK Biobank data through an application process: http://www.ukbiobank.ac.uk/register-apply/. The registration is then reviewed by the Access Management Team of the UK Biobank. Genome-wide association studies summary statistics results are publicly available: http://www.nealelab.is/blog/2017/7/19/rapid-gwas-of-thousands-of-phenotypes-for-337000-samples-in-the-uk-biobank  Model definition files are described in Gamazon et al. 2015. Code for the following analyses is publicly available: PrediXcan: https://github.com/hakyimlab/PrediXcan. S-PrediXcan: https://github.com/hakyimlab/MetaXcan.

The following previously published datasets were used:

| Author(s) | Year | Dataset title | Dataset URL | Database and Identifier |
|---|---|---|---|---|
| Nikpey M, Goel A, Won H, Hall LM, Willenborg C, Kanoni S, Saleheen D | 2015 | Coronary ARtery DIsease Genome wide Replication and Meta-analysis (CARDIoGRAM) plus The Coronary Artery Disease (C4D) Genetics (CARDIoGRAMplusC4D) | ftp://ftp.sanger.ac.uk/pub/cardiogramplusc4d/2015/mi.additive.Oct2015.pub.zip | CARDIoGRAMplusC4D, mi.additive.Oct2015 |
| The GTEx Consortium | 2017 | GTEx Port | https://www.ncbi.nlm.nih.gov/projects/gap/cgi-bin/study.cgi?study_id=phs000424.v6.p1 | NCBI dbGaP, phs000424.v6.p1 |
| Locke AE, Kahali B, Berndt SI, Justice AE, Pers TH, Day FR | 2015 | Genetic studies of body mass index yield new insights for obesity biology | http://www.broadinstitute.org/collaboration/giant/images/f/f0/All_ancestries_SNP_gwas_mc_merge_nogc.tbl.uniq.gz | Broad Institute, All_ancestries_SNP_gwas_mc_merge_nogc.tbl |
| Westra H-J, Peters MJ, Esko T, Yaghootkar H, Schurmann C, Kettunen J | 2013 | Systematic identification of trans eQTLs as putative drivers of known disease associations | https://genenetwork.nl/bloodeqtlbrowser/2012-12-21-CisAssociationsProbeLevelFDR0.5.zip | Gene Network, 2012-12-21-CisAssociationsProbeLevelFDR0.5 |
| Stitziel NO, Stirrups KE, Masca NGD, Erdmann J, Ferrario PG, Konig IR | 2016 | Coding Variation in ANGPTL4, LPL, and SVEP1 and the Risk of Coronary Disease | http://www.cardiogramplusc4d.org/media/cardiogramplusc4d-consortium/data-downloads/MICAD.EUR.ExA.Consor- | CARDIoGRAMplusC4D, MICAD.EUR.ExA.Consortium.PublicRelease.310517 |

| | | | tium.PublicRelease. 310517.zip | |
|---|---|---|---|---|
| Nielsen JB, Thorolfsdottir RB | 2018 | Biobank-driven genomic discovery yields new insight into atrial fibrillation biology | http://csg.sph.umich. edu/willer/public/ afib2018/nielsen-thorolfsdottir-willer-NG2018-AFib-gwas-summary-statistics.tbl.gz | University of Michigan, nielsen-thorolfsdottir-willer-NG2018-AFib-gwas-summary-statistics.tbl |

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
