## [Decision Letter]

[Editors’ note: a previous version of this study was rejected after peer review, but the authors submitted for reconsideration. The first decision letter after peer review is shown below.]

Thank you for submitting your work entitled "Bid maintains mitochondrial cristae in mice and protects against human cardiac disease in an integrative genomics study" for consideration by *eLife*. Your article has been reviewed a Senior Editor, a Reviewing Editor, and three reviewers.

Our decision has been reached after consultation between the reviewers. Based on these discussions and the individual reviews below, we regret to inform you that your work will not be considered further for publication in *eLife*.

As you will see reading the specific comments below that there is interest in your work and especially the aspects regarding human genomics. However, the several serious issues raised by the reviewers are too substantial for revision within the two month revision timeline that *eLife* espouses. We felt that some aspects of the results were not convincingly demonstrated and that others used too few animals for rigorous conclusions. One other key issue that we felt needed more attention was the link between the alterations in cristae seen in cells and animal models and the data collected in humans. If you are able to address all the issues noted and/or delete aspects with marginal effects, *eLife* would be prepared to reconsider this work, but only as a new submission and with no commitments to take it forward.

*Reviewer #1:*

In Figure 1 the authors mention metabolic stress but only test quite extreme cell stresses such as serum withdrawal. To better support the claims a more physiological stress such as shifting the cells into galactose instead of glucose medium to drive Ox/Phos is recommended. The differences in Figure 1I with vehicle and digitonin between cells with and without Bid may be statistically significant but is not compelling.

The EM data in Figure 2A is interesting. It would be important to validate this by examining cells in tissue fixed and taken from mice directly without culturing. Is there something special about myeloid precursor cells or is Bid regulating cristae morphology in other cell types? In particular the authors should examine the heart (with and without epinephrine treatment) as this is the focus later on in the manuscript. It is strange that Puma and Bim expression is decreased in Bid null cells (Figure 2—figure supplement 2C). This reveals that pleiotropic effects in these cells may be yielding the phenotypes indirectly of or far removed from Bid expression itself.

Figure supplement 1D is inadequate. Higher magnification is needed. Multiple cell images need to be shown. Cytosolic Bid will undoubtedly overlap with mitochondria to some extent. FRAP to remove the cytosolic pool of Bid may reveal more compelling mitochondrial localization of remaining Bid. Figure 3A is better and could replace Figure supplement 1D. How does Bid bind mitochondria? Is it binding MTCH2? If this accounts for the localization then how does this matter considering the PrediXcan data showing no link with MTCH2 (but an effect of BID) in Figure 6B?

How Bid regulates cristae morphology remains obscure and limiting the significance of this study. The authors find a decrease in Complex I activity in Bid KO heart tissue and lower ATP synthase dimers that could explain the cristae morphology change. But equally possible is that the change in cristae morphology may mediate the Complex V change. Thus, only a correlation is shown and no mechanism revealed.

Does M148T Bid rescue cristae morphology in Bid KO cells? MPCs or heart?

Overall, there is a lot of data and Bid KO mitochondria do seem different than WT. But how this occurs, and the significance of the changes is not clear. Whether the mitochondrial morphology and biochemical changes account for the results in the population studies is also not clear. The one take home lesson is that Bid appears to have functions above and beyond apoptosis regulation, but what these are is unknown.

*Reviewer #2:*

This manuscript that provides interesting data showing that full length Bid modulates mitochondrial cristae and respiration under homeostatic conditions independently of caspase-8 cleavage (D59A) and its BH3-domain. Additional data using a gene-based approach applied to a biobank, validated in an independent largescale GWAS, decreased BID gene expression correlates with myocardial infarction (MI). Carrier status with nonsynonymous variation, including M148T, in the membrane binding domain of Bid also associates with MI predisposition.

1) The Introduction mentions that there is an association between Bid expression and MI. Could it just be that MI leads to the loss of Bid, rather than BID regulating MI?

2) Subsection “*Bid -/-* cells have a cristae defect that can be rescued with BH3-mutated or D59-mutated Bid” mention that subcellular fractionation show that Bid is in the mitochondrial fraction. Were controls done to check for contaminations from other fractions. Were the mitochondria treated with a protease to determine if Bid was attached to the mitochondria as opposed to being in the mitochondrial matrix. These are important controls.

3) It is concluded that acute epinephrine (30μg) treatment leads to an increase in LVIDd and LVIDs. It would be useful to provide a time course for the changes. The n's are also low for these studies.

4) Data are presented showing that 18 hours after a 0.5 mg/kg dose of epinephrine there is an increase in LVIDd and LVIDs. This also resulted in a decrease in FS and EF. Does this decrease in EF and FS persist or does it recover? More detailed characterization is needed.

5) The description of the PrediXcan analysis and some of the other bioinformatic analyses is a bit long and could be shortened.

6) N's for the supercomplex data are low (n=3). I don't find the data in Figure 3F convincing, particularly given the low n. Also, the Figure 2—figure supplement 2, doesn't show any significant differences in any of the complex activity. Also subsection “Complex I containing supercomplexes display decreased activity in *Bid-/-* cardiac mitochondria” says there is a significant decrease in activity of complex I within the SC (Figure 3F). However, Figure 3F looks at band density, not activity. Activity is measured in the supplemental figure, but it doesn't show a change in activity.

*Reviewer #3:*

Salsbury-Ruf et al., report that Bid loss results in increased cell death (contrary to known role of Bid), mitochondrial cristae defects, and decreased respiration at the cellular level. In knockout mice, they find impaired heart function when challenged with drugs. They find evidence, at the human level, that lower genetically determined levels of BID are associated with increase myocardial infarction using their biobank data (BioVU). Finally, they find that excess variation in the coding region of BID also associates with myocardial infarction.

The results in cells and mice seems compelling. I'll defer the review of the experimental setup and implementation to other reviewers more familiar with the techniques and model systems. Reading through, I did not identify any obvious statistical problems with the analysis of this first section.

The association between BID and myocardial infarction risk in the Vanderbilt's electronic health record based Biovu sounds also compelling. However, the rationale behind the choice of the 9 tissues out of 50 in GTEx was less clear. The figures show p values but the tissue is not clearly stated. Each tissue model yields a different p values. The rationale of the choice of tissues should be explained and the figures should clearly label which tissue result is being presented.

Heritability analysis is performed to pre-filter genes to be investigated. However, prediction models from different tissues are used in the analysis. The latter decision could be justifiable if the prediction models yielded cross validated significance, which is evidence of genetic basis of the gene expression trait. In this case, the pre-filtering of genes by heritability in different tissues would be unnecessary.

It is good to see that BID was associated with at least modest significance in heart tissue in CARDIOGRAM. But how about the other tissues? Were they not significant?

The much larger sample size in the CARDIOGRAM study did not result in improved significance, which seems a bit concerning. The argument of homogeneity of ethnicity in BioVU vs CARDIOGRAM sounds somewhat plausible but not completely convincing. GTEx samples are not as homogeneous as implied in the text.

It is reassuring to see replication in the UK Biobank data. However, why are only 3 tissues shown in Figure 7 C (skin, nerve-tibial, brain-cerebellum)? Were they selected for being significant?

Clarification of these points are needed to make it clear that there was no cherry picking of the tissues and associations presented.

[Editors’ note: what now follows is the decision letter after the authors submitted for further consideration.]

Thank you for resubmitting your work entitled "Bid maintains mitochondrial cristae structure and protects against cardiac disease in an integrative genomics study" for further consideration at *eLife*. Your revised article has been favorably evaluated by Mark McCarthy (Senior Editor), a Reviewing Editor, and three reviewers.

The manuscript has been improved but there are some remaining issues that need to be addressed before acceptance, as outlined below:

*Reviewer #2:*

Results section

I find the sentence about the possibility of integrating the results in clinical application unclear and vague.

Subsection “PrediXcan analysis reveals decreased BID expression associates with myocardial infarction”. Why is the heritability of BECN1 consistent with a known protective role of autophagy and cardiovascular disease?

Subsection “Validation in BioVU and CARDIoGRAMplusC4D GWAS” I don't find the heart tissue as mentioned here: "In a separate BioVU sample set (see Materials and methods section and Figure 7—figure supplement 6c and d), we observed a significant correlation (p=0.002) between decreased genetically determined BID expression in heart tissue and MI."

---

## [Author Response]

[Editors’ note: the author responses to the first round of peer review follow.]

Reviewer #1:In Figure 1 the authors mention metabolic stress but only test quite extreme cell stresses such as serum withdrawal. To better support the claims a more physiological stress such as shifting the cells into galactose instead of glucose medium to drive Ox/Phos is recommended. The differences in Figure 1I with vehicle and digitonin between cells with and without Bid may be statistically significant but is not compelling.

Given the focus of the manuscript is on heart metabolism, which we characterize with specific respiratory complex substrates (now with increased numbers of mice, Figure 6E and F), we have removed this data from the manuscript.

We agree that serum withdrawal is indeed an extreme cell stress. We find that myeloid progenitor cells are very metabolically plastic (reviewed in Hsu and Qu, (2013)) and can utilize multiple carbon sources including galactose in the presence of 10% serum. It has also been shown that galactose does not necessarily drive mitochondrial metabolism (Elkalaf et al., 2013) and its impact on mitochondrial metabolism is cell type specific and thus the metabolic status of myeloid cells is no longer relevant to the revised manuscript.

Regarding the cytochrome c mobilization assay, we agree that this point, while supportive of a cristae defect, is not a major observation and has been removed from this version of the manuscript to keep the results focused.

Of note, we did not anticipate that there would be a large difference using digitonin. To release a substantial amount of cytochrome c, apoptotic tBid must be used. (Scorrano et al., 2002).

The EM data in Figure 2A is interesting. It would be important to validate this by examining cells in tissue fixed and taken from mice directly without culturing. Is there something special about myeloid precursor cells or is Bid regulating cristae morphology in other cell types? In particular the authors should examine the heart (with and without epinephrine treatment) as this is the focus later on in the manuscript.

We have now analyzed heart mitochondria by EM in both WT and *Bid -/-* mice in the presence and absence of an Epinephrine challenge. We have incorporated this new data into Figure 4A and B. We have made several major observations.

There is a clear loss of organization of mitochondria within the *Bid -/-* cardiac tissue. Whereas mitochondria in WT cardiac tissue are localized to areas between linear cardiomyocytes, mitochondria within the *Bid-/-* cardiac tissue are found scattered throughout the tissue in a discontinuous and disorganized manner.

Secondly, there is a striking difference in cristae morphology both before and after Epinephrine challenge (18-hour challenge at 0.5 mg/kg). We have assessed this with measurement of cristae widths. We find that in untreated tissue, *Bid-/-* mitochondria are generally less electron dense, and more rounded in shape. After treatment, *Bid-/-* mitochondria become increasing less electron dense and damaged. WT mitochondria, however, are lamellar in shape and are not significantly altered by Epinephrine treatment. We have now provided strong

evidence for a role for Bid in cristae morphology both in MPCs and tissue taken directly from WT and *Bid-/-* mice.

It is strange that Puma and Bim expression is decreased in Bid null cells (Figure 2—figure supplement 2C). This reveals that pleiotropic effects in these cells may be yielding the phenotypes indirectly of or far removed from Bid expression itself.

Regarding the other BH3-only proteins, we agree it is intriguing that there is decreased expression of Bim, Puma, and Bad in *Bid -/-* myeloid progenitors. Examination of left ventricular cardiomyocytes for these proteins also reveals decreased expression of Bad and Bim (Puma was not observed in either WT or *Bid-/-* mice). These blots have been added to Figure 2—figure supplement 2C.

While loss of these proteins may have pleiotropic effects beyond the scope of this manuscript, this data demonstrates that the cristae abnormalities observed in *Bid-/-* cells and tissue are not due to a compensatory overexpression of other BH3-only proteins. Furthermore, we are able to restore normal mitochondrial morphology by adding back non-apoptotic Bid to *Bid -/-* cells. Lastly, we now have new genetic evidence for a role for BID expression and cardiac diseases. Patients clustered by gene expression reveal that those which fall within the lowest 5% of BID expression are enriched for cardiac phenotypes, including myocardial infarction, on average, >4-fold more than MI patients within the Vanderbilt University’s Synthetic Derivative.

Figure supplement 1D is inadequate. Higher magnification is needed. Multiple cell images need to be shown. Cytosolic Bid will undoubtedly overlap with mitochondria to some extent. FRAP to remove the cytosolic pool of Bid may reveal more compelling mitochondrial localization of remaining Bid. Figure 3A is better and could replace Figure supplement 1D. How does Bid bind mitochondria? Is it binding MTCH2? If this accounts for the localization then how does this matter considering the PrediXcan data showing no link with MTCH 2(but an effect of BID) in Figure 6B?

We have taken a biochemical approach to definitively demonstrate full-length Bid mitochondrial localization. In Figure 3 we now include the following results:

Isolation of crude and Percoll purified mitochondria from liver (to remove light membrane contamination), reveals the localization of full-length Bid in the absence of apoptosis (Figure 3B).

Proteinase K (PK) treatment of isolated liver mitochondria validate the submitochondrial localization of Bid by an independent method. These results also reveal Bid at the mitochondria even after proteinase K treatment in conditions in which we observe degradation of Bak (Figure 3C).

Using a sub mitochondrial fractionation technique that takes advantage of the sensitivity of the mitochondria to hypotonic stress, we are able to isolate the outer membrane (OMM) from the mitoplast (inner membrane and matrix). We demonstrate Bid in both the OMM as well as in the mitoplast fraction, consistent with our PK results. Thus, we show homeostatic localization of Bid to the mitochondria and find a pool of Bid in the matrix, in position to exert a role in cristae structure.

In regard to MTCH2, while there is literature supporting a role for MTCH2 as a docking platform for apoptotic tBid (Zaltsman et al., 2010) it remains to be determined if full-length Bid (or phosphorylated mutant Bid^S61AS78A (AA)^ which is thought to be mitochondrially localized) associates with MTCH2. Furthermore, it is known that Bid also associates with the mitochondrial lipid cardiolipin as a docking platform (Kim et al., 2004). We propose that Bid’s ability to stabilize cristae is dependent in part, on its matrix localization and would argue that MTCH2 is not essential for this process.

In the literature, we find that studies on MTCH2 by the Gross lab demonstrate that both Bid^AA^ and MTCH2^F/F^ Vav1-Cre+ mouse models show loss of hematopoietic stem cell (HSC quiescence), and from this data they conclude that by extension, both models would have comparable changes in mitochondria (Maryanovich et al., 2015). While this is possible, there is no direct evidence that the structural changes observed in mitochondria are mediated through an interaction of Bid with MTCH2. Indeed, there is no data that demonstrates a non-apoptotic function of the Bid/MTCH2 interaction. Consequently, no structure- function studies have been performed, and thus the minimal binding domain for the interaction between Bid and MTCH2 is unknown.

Lastly, as noted by the reviewer, we find no genetic evidence for a role for MTCH2 both at the level of heritability as well as any association with expression and myocardial infarction. We do however find an association between increased MTCH2 gene expression and increased BMI, which is supported by data previously found in the literature (Bar-Lev et al., 2016).

Taking these observations into consideration, as well as our new result revealing Bid in the matrix (as well as associated with MCl-1 discussed below), we have de-emphasized the discussion of MTCH2 throughout the manuscript.

How Bid regulates cristae morphology remains obscure and limiting the significance of this study. The authors find a decrease in Complex I activity in Bid KO heart tissue and lower ATP synthase dimers that could explain the cristae morphology change. But equally possible is that the change in cristae morphology may mediate the Complex V change. Thus, only a correlation is shown and no mechanism revealed.

In light of our new data suggesting Bid can localize to the mitochondrial matrix, we tested whether Bid could interact with the BCl-2 protein MCl-1. MCl-1 has been demonstrated to localize to the mitochondrial matrix and impact cristae structure as well as cardiac function including cardiomyopathy (Wang et al., 2012 and Thomas et al., 2013).

We present data demonstrating co-immunoprecipitation of WT-Bid (FlagHA tagged-Bid) with WT, outer membrane (MCl-1^OM^), and matrix localizing (MCl-1^Matrix^) forms of MCl-1. This supports a potential role for Bid within the mitochondrial matrix through MCl-1 (Figure 9a). Bid’s interaction with MCl-1^Matrix^ is especially interesting as another BH3-only protein, Bim, does not interact with matrix MCl-1 (Perciavalle et al., 2012). We propose that loss of Bid may be indirectly altering the function, sub-mitochondrial localization or protein-protein interactions of MCl-1, to impact cristae structure.

We further show that M148T mutated Bid, (our loss of function SNP revealed in BioVU patients) has decreased interaction with MCl-1^Matrix^ compared to both WT and Bid^D59A^ (Figure 9B). This corresponds to a loss of Bid-mediated cristae structure in the presence of Bid^M148T^ (new data), providing further evidence for a role for Bid’s helix 6 in regulating cristae structure through association with MCl-1.

Does M148T Bid rescue cristae morphology in Bid KO cells? MPCs or heart?

We show genetically that in a BioVU cohort, carrier status of the Bid SNP M148T increases association with MI. When this point mutation is made and combine with BH3 mutated Bid (rescue mutant) we are unable to restore MPC respiration.

Consistent with this finding, new EM data from MPCs (Figure 8E-G) reveals that this double mutant results in significantly decreased numbers of cristae compared to Bid^BH3^ alone (in *Bid-/-* cells), revealing that Bid^M148T^ results in a loss of mitochondrial structure in addition to function previously shown. This is also supported by our findings for MCl-1.

Overall, there is a lot of data and Bid KO mitochondria do seem different than WT. But how this occurs, and the significance of the changes is not clear. Whether the mitochondrial morphology and biochemical changes account for the results in the population studies is also not clear. The one take home lesson is that Bid appears to have functions above and beyond apoptosis regulation, but what these are is unknown.

We have now provided additional data to address the potential biological mechanism for the role of full-length Bid at the mitochondria. In addition, we can now more strongly connect the mouse model to human genetics both with additional experimental data as well as a new genetic approach.

We have expanded our echocardiogram analysis of WT and *Bid-/-* mice. We now have additional data at baseline (without treatment), 18 hours after Epinephrine, and at a new recovery time point (5 days after treatment). This data reveals decreased ejection fraction as well as increased end diastolic and end systolic volumes, parameters which are observed in the setting of MI and directly linked to poor patient prognosis after an MI (Di Bella et al., 2013 and White et al., 1987) (Figure 4C-G).

We have also evaluated fibrosis in our model. Following myocardial damage such as seen in MI, the myocardium undergoes remodeling to repair damage. This remodeling manifests as fibrosis, and fibrosis can be used as a measure of myocardial damage. (Talman et al., 2016). Using Trichrome stain, we determined the amount of fibrosis both at the 18 hour and recovery time points using an unbiased quantitative algorithm (See Materials and methods section). We find increased fibrotic damage in the *Bid-/-* mice compared to WT mice, directly linking our results to observations in patients (Figure 5).

We now find Bid in the mitochondrial matrix and reveal an association between Bid and the mitochondrial matrix form of MCl-1, the other BCl-2 family member reported to localize to the mitochondrial matrix and to impact cristae structure. We find that this association is partially dependent on an intact Bid helix 6, thus implicating an association with MCl-1^Matrix^ in Bid’s role to regulate cristae structure.

We used a genetic approach to determine if there is an association between 41 genes known to be involved in the maintenance of cristae structure and myocardial infarction. PrediXcan analysis of these genes revealed decreased expression of two genes associated with MI; MCL-1 and MTX1 (Metaxin-1), a mitochondrial transporter known to associate with the mitochondrial contact site and cristae organizing system (MICOS). Thus, we have implicated MCl-1 both as an interacting partner with Bid as well as associated with MI at the genetic level. We have further implicated mitochondrial cristae structure in susceptibility to MI.

We also took advantage of the BioVU Synthetic Derivative, which contains with over 2.8 million de-identified patient records. We constructed a contingency table to determine the relative risk of MI with cardiomyopathy, as cardiac specific loss of MCl-1 results in cardiomyopathy (Wang et al., 2012 and Thomas et al., 2013). We find a very significant relative risk score for this association compared to known risk factors for MI such as hypertension (Figure 9Dd. Thus, implicating a cardiac disease with a mitochondrial etiology as a significant risk factor for MI.

Lastly, as mentioned above, we have also substantially strengthened the connection between BID expression and data from cells and mice. In particular, we have identified the patients in BioVU with the lowest 5% of BID expression, mimicking a phenotype of a genetic knockout as closely as possible. We find that MI, is >4-fold higher in these individuals versus patients within the rest of the Vanderbilt Synthetic Derivative. This new data substantially strengthens the association between our mouse and human data.

Reviewer #2:This manuscript that provides interesting data showing that full length Bid modulates mitochondrial cristae and respiration under homeostatic conditions independently of caspase-8 cleavage (D59A) and its BH3-domain. Additional data using a gene-based approach applied to a biobank, validated in an independent largescale GWAS, decreased BID gene expression correlates with myocardial infarction (MI). Carrier status with nonsynonymous variation, including M148T, in the membrane binding domain of Bid also associates with MI predisposition.1) The Introduction mentions that there is an association between Bid expression and MI. Could it just be that MI leads to the loss of Bid, rather than BID regulating MI?

The approach that we have taken with PrediXcan directly addresses this valid concern. PrediXcan analyzes the effects of germline alterations, such as polymorphisms, on gene expression. We exclude both environmental effects as well as effects the disease itself could have on the expression of the gene. These imputed values of genetically determined expression are based on a reference transcriptome such as the GenotypeTissue Expression (GTEx) project which genotypes tissue donors and links this to genotyping to gene expression (mRNA). We find that when PrediXcan is applied to BioVU as well as multiple replication cohorts, decreased genetic component of BID gene expression is associated with MI, suggesting the direction is from BID to MI.

2) Subsection “Bid -/- cells have a cristae defect that can be rescued with BH3-mutated or D59-mutated Bid” mention that subcellular fractionation show that Bid is in the mitochondrial fraction. Were controls done to check for contaminations from other fractions. Were the mitochondria treated with a protease to determine if Bid was attached to the mitochondria as opposed to being in the mitochondrial matrix. These are important controls.

We have significantly increased our biochemical analysis of Bid at the mitochondria. In a revised Figure 3 we now provide several lines of additional evidence.

These include:

Both crude and Percoll purified mitochondria isolated from mouse liver (absent of light membrane contamination) reveals the presence of full-length Bid (Figure 3B).

Proteinase K treatment of mitochondria showing the presence of Bid in the protease treated fraction in conditions in which Bak is removed (degraded) (Figure 3C).

Sub-mitochondrial fractionation using an osmotic shrink-swell approach separating the outer membrane (OMM) from the mitoplast (inner membrane and matrix). We find a fraction of Bid present in the mitoplast enriched fraction.

Lastly, we co-immunoprecipitated FlagHA-tagged Bid with WT, outer mitochondrial membrane, and matrix forms of the BCl-2 family member MCl-1. Mcl1 has been shown to play a role in mitochondrial cristae structure (Perciavalle et al., 2012) and cardiac dysfunction (Wang et al., 2012 and Thomas et al., 2013). Furthermore, we find that Bid^M148T^ has decreased interaction with MCl-1^Matrix^ compared to WT or Bid^D59A^.

This additional evidence suggests that there is indeed a fraction of full-length Bid that is present not only on the outer mitochondrial membrane but also within the matrix of the mitochondria. Bid can associate with the matrix form of MCl-1 and is involved in the regulation of mitochondrial cristae.

3) It is concluded that acute epinephrine (30μg) treatment leads to an increase in LVIDd and LVIDs. It would be useful to provide a time course for the changes. The n's are also low for these studies.

While we do see an increase in LVIDd/s with an acute dose of 30μg of Epinephrine, we do not see a decrease function in other cardiac parameters likely as this damage needs time to manifest. We agree that these data could be improved with more replicates. To best strengthen these results, we have focused on acquiring additional replicates with measurements taken at baseline, 18 hours after epinephrine (at 0.5 mg/kg), as well as 5 days after epinephrine to evaluate recovery.

We have now added significantly more mice for a complete analysis before epinephrine WT n=12 and *Bid-/-* n=12), at the 18-hour time point (WT n=12 and *Bid-/-* n=11), and a recovery timepoint 5 days later (WT n=5 and *Bid-/-* n=6). At 18 hours post-Epi, we see an increase in LVIDd as well as LVIDs as well as a decrease in EF and a trend for a decrease in FS (p=0.068) for *Bid-/-* mice. Furthermore, we observe increased end diastolic and end systolic volumes. These parameters are directly linked to poor patient prognosis after an MI (Di Bella et al., 2013 and White et al., 1987) (Figure 4C-G).

4) Data are presented showing that 18 hours after a 0.5 mg/kg dose of epinephrine there is an increase in LVIDd and LVIDs. This also resulted in a decrease in FS and EF. Does this decrease in EF and FS persist or does it recover? More detailed characterization is needed.

Interestingly, at 5 days, the recovery time point, we observe that *Bid*-/- mice functionally return to baseline parameter and are not different compared to WT mice. We were interested to know however, if similar to fibrotic damage observed in patients after an MI (Talman et al., 2016), we would observe increased fibrosis in the *Bid-/-* hearts after Epinephrine. Using Trichrome stain, we measured the amount of fibrosis both at the 18-hour time point and 5 days later using a quantitative algorithm. We find increased fibrotic damage in the *Bid*-/- mice compared to WT mice at both time points, directly linking our results to observations in MI patients (Figure 5).

5) The description of the PrediXcan analysis and some of the other bioinformatic analyses is a bit long and could be shortened.

We appreciate that the genetics analysis could be described more concisely. We have addressed this concern and have substantially revised this section to emphasize our most important findings. We combined the exome-level analysis into the site-directed mutagenesis section to make the genetics section more concise and to emphasize the functional validation component of our studies. We further combined the other BCl-2 family analysis into primary PrediXcan analysis to emphasize the unique role of Bid. Additional findings for the BCl-2 family are now in a new supplementary information section.

6) N's for the supercomplex data are low (n=3). I don't find the data in Figure 3F convincing, particularly given the low n. Also, the Figure 2—figure supplement 2, doesn't show any significant differences in any of the complex activity. Also, subsection “Complex I containing supercomplexes display decreased activity in Bid-/- cardiac mitochondria” says there is a significant decrease in activity of complex I within the SC (Figure 3F). However, Figure 3F looks at band density, not activity. Activity is measured in the supplemental figure, but it doesn't show a change in activity.

Regarding the number of runs for this assay, in particular CI, we should clarify that the data presented are representative of 4 independent experiments. However, we present quantitation from one representative experiment (mitochondrial isolated from 3 WT and 3 *Bid-/-* mice). As enzymatic activity can be impacted by small changes in temperature and time in which the enzyme is exposed to substrate, our analysis is done on mitochondria run on one gel for accuracy and precision.

All of data originally presented in Figure 3 as well as in Figure 2—figure supplement 2 measures enzymatic activity. We appreciate the original labeling is not clear. To address this, we have clarified the labeling.

In our original Figure 3. we presented in-gel activity (IGA) results for respiratory complex I and ATP synthase (Complex V) from isolated heart mitochondria. Each complex tested has a Coomassie stained gel side by side with the corresponding activity assay. The activity assays are based on a colorimetric change that occurs when the active enzyme in the gel is incubated in the presence of its corresponding substrate (NADH for CI and ATP for CV and cytochrome c for CIV). We then measure band density as a read-out of activity as the more active the enzyme, the more robust the color change and therefore the denser the band. All in-gel activity assays measure enzymatic activity by determining band density.

We also focused our results on ATP synthase activity. In-gel activity data is now supported by additional replicates of real-time ATP synthesis analysis from heart tissue, and these data are now presented together in Figure 6. In-gel activity analysis for CI and CIV are now in Figure 6—figure supplement 1.

Reviewer #3:Salsbury-Ruf et al., report that Bid loss results in increased cell death (contrary to known role of Bid), mitochondrial cristae defects, and decreased respiration at the cellular level. In knockout mice, they find impaired heart function when challenged with drugs. They find evidence, at the human level, that lower genetically determined levels of BID are associated with increase myocardial infarction using their biobank data (BioVU). Finally, they find that excess variation in the coding region of BID also associates with myocardial infarction.The results in cells and mice seems compelling. I'll defer the review of the experimental setup and implementation to other reviewers more familiar with the techniques and model systems. Reading through, I did not identify any obvious statistical problems with the analysis of this first section.The association between BID and myocardial infarction risk in the Vanderbilt's electronic health record based Biovu sounds also compelling. However, the rationale behind the choice of the 9 tissues out of 50 in GTEx was less clear. The figures show p values but the tissue is not clearly stated. Each tissue model yields a different p values. The rationale of the choice of tissues should be explained and the figures should clearly label which tissue result is being presented.

We have now reorganized the presentation of our discovery results and replication to focus on the association in the most relevant tissue, heart (for completeness, results in non-heart tissues were moved to a new supplementary information section).

Our original analyses were conducted in the 9 tissues. It’s important to point out that there is quite a bit of shared genetic architecture of gene expression between tissues (GTEx Consortium, 2017), and thus it’s possible that even if the “causal” or “pathogenic” tissues or cell types are not available, we could leverage this shared architecture to identify genetic associations with disease (Gamazon et al., 2018). And of course, not all tissue is likely to be causal tissues for our disease of interest. It’s for example, unclear whether some of the reproductive tissues in GTEx, are relevant to MI. Finally, not all tissues have prediction models with sufficiently high quality due to sample size for the tissues and/or due to heterogeneity in genetic effect sizes. However, we do observe the association with MI in heart tissue.

Heritability analysis is performed to pre-filter genes to be investigated. However, prediction models from different tissues are used in the analysis. The latter decision could be justifiable if the prediction models yielded cross validated significance, which is evidence of genetic basis of the gene expression trait. In this case, the pre-filtering of genes by heritability in different tissues would be unnecessary.

We thank the reviewer for the opportunity to clarify this analysis. The heritability estimates we provided (from DGN) are consistent with the imputation R2 from the prediction models (in GTEx heart tissue). Pre-filtering based on heritability or imputation quality would yield the same results in this case. We opted to report the heritability in DGN because of its much larger sample size (n=922). We have clarified this in the text as follows: “Furthermore, consistent with the heritability estimates in DGN, the expression of MTCH2 in heart could not be imputed well using genetic variation from the cis region of the gene […]”.

It is good to see that BID was associated with at least modest significance in heart tissue in CARDIOGRAM. But how about the other tissues? Were they not significant?

We think heart is the most relevant tissue of all GTEx tissues examined. We

have now changed the presentation to highlight this tissue. We did incorporate the other tissues for completeness. We emphasize that we have considerable prior evidence for BID’s role in heart disease from our functional studies (in mice and cells), which are now highlighted in the workflow (Figure 1). We are not attempting an unbiased interrogation of the genome in search of disease-associated genes. Rather, we are replicating, in human studies, what we significantly observe in mice and cells. Furthermore, our study attempts to elucidate the mechanism for BID’s role in human cardiac disease and in regulating cristae structure.

The much larger sample size in the CARDIOGRAM study did not result in improved significance, which seems a bit concerning. The argument of homogeneity of ethnicity in BioVU vs CARDIOGRAM sounds somewhat plausible but not completely convincing. GTEx samples are not as homogeneous as implied in the text.

We believe that the heterogeneity of ethnicity can have a substantial impact on the results. In fact, the proportion of patients of non-European descent in the

CARDIOGRAM dataset (23%) is substantially higher than the proportion in GTEx (10%). We further emphasize that our study was not an unbiased search for an MI-associated gene, however it was motivated by our extensive functional studies of Bid in mice, that have already demonstrated stress-induced fibrosis, myocardial dysfunction, and mitochondrial structural perturbations.

It is reassuring to see replication in the UK Biobank data. However, why are only 3 tissues shown in Figure 7 C (skin, nerve-tibial, brain-cerebellum)? Were they selected for being significant?

We have now modified the presentation of our results. We now focus on the tissue of relevance, heart. For completeness, we present the results in non-heart tissues and have moved these results to the supplementary information and supplemental Figure 7—figure supplement 3C.

Clarification of these points are needed to make it clear that there was no cherry picking of the tissues and associations presented.

Please see above. Since we have now focused our results on the most relevant tissue, heart, it is now clear that the additional non-heart tissues were included only for completeness. Notably, our conclusions do not depend on the inclusion of these non-heart tissues.

[Editors' note: the author responses to the re-review follow.]

The manuscript has been improved but there are some remaining issues that need to be addressed before acceptance, as outlined below:Reviewer #2:Results sectionI find the sentence about the possibility of integrating the results in clinical application unclear and vague.

We appreciate how this sentence was not clear regarding the potential clinical applications for our findings. Our intent was to propose the idea that the genetic approach we used here, specifically PrediXcan, could be used as a way to clinically test predisposition to disease. PrediXcan analysis is based on a reference transcriptome that can be applied to a biobank or GWAS dataset. This transcriptome is based upon SNPs from healthy individuals, and these SNPs have a high allele frequency within the general population. Thus, it is likely that the risk alleles at these SNPs will often be found in patients. If a particular combination of SNPs results in changes in expression of a gene associated with a disease, then the effect on disease risk could be estimated and consequently in some cases patient care may reflect this potential risk.

However, we do feel that this statement could be reaching as a prospective study testing this idea may be the next appropriate step in the integration of this type of analysis into the clinic. We have removed this sentence from the Introduction in light of this and appreciate that this type of broad application has yet to be determined.Subsection “PrediXcan analysis reveals decreased BID expression associates with myocardial infarction”. Why is the heritability of BECN1 consistent with a known protective role of autophagy and cardiovascular disease?

The manuscript no longer makes this connection. We have removed this from Subsection “PrediXcan analysis reveals decreased BID expression associates with myocardial infarction”, in the current version.

Our heritability analysis revealed that in addition to BID, BECN1 was significantly heritable, and thus like BID also under genetic control. We originally stated that we found this to be interesting as it is “consistent with a protective role of autophagy and cardiovascular disease.” We completely agree with the reviewer that heritability alone does not connect these two observations.

Our original intent was to convey the idea that there is a previously known connection between Beclin1’s role in autophagy and protection against cardiovascular disease. This is unique for Beclin1 among the proteins assessed in our secondary analysis, as there are previously no known roles (to the best of our knowledge) for these proteins in the protection against heart disease or myocardial infarction. We agree however that heritability does not reflect this role for Beclin1.

In fact, PrediXcan analysis of BECN1 (Figure 7—figure supplement 7A) reveals an association with heart phenotypes. We find that significantly increased genetically-determined expression associates with heart failure, while decreased expression associates with cardiac shunt and primary cardiomyopathy. Thus, the association of BECN1 with heart failure is likely to be driven by genetic variation.

Subsection “Validation in BioVU and CARDIoGRAMplusC4D GWAS” I don't find the heart tissue as mentioned here: "In a separate BioVU sample set (see Materials and methods section and Figure 7—figure supplement 6c and d), we observed a significant correlation (p=0.002) between decreased genetically determined BID expression in heart tissue and MI."

We appreciate the opportunity to clarify our genetic data presented in Figure 7—figure supplement 6C and D. We find a significant correlation with decreased BID expression (p=0,002) with MI in aorta. We have clarified this in Subsection “Validation in BioVU and CARDIoGRAMplusC4D GWAS” in the present draft by indicating that the association was found in aorta.